# FIC-TSC: Learning Time Series Classification with Fisher Information Constraint

**Xiwen Chen** [1]  **Wenhui Zhu** [2]  **Peijie Qiu** [3]  **Hao Wang** [1]  **Huayu Li** [4]
**Zihan Li** [5]  **Yalin Wang** [2]  **Aristeidis Sotiras** [3]  **Abolfazl Razi** [1]

## Abstract

Analyzing time series data is crucial to a wide spectrum of applications, including economics, online marketplaces, and human healthcare. In particular, time series classification plays an indispensable role in segmenting different phases in stock markets, predicting customer behavior, and classifying worker actions and engagement levels. These aspects contribute significantly to the advancement of automated decision-making and system optimization in real-world applications. However, there is a large consensus that time series data often suffers from domain shifts between training and test sets, which dramatically degrades the classification performance. Despite the success of (reversible) instance normalization in handling the domain shifts for time series regression tasks, its performance in classification is unsatisfactory. In this paper, we propose *FIC-TSC*, a training framework for time series classification that leverages Fisher information as the constraint. We theoretically and empirically show this is an efficient and effective solution to guide the model converge toward flatter minima, which enhances its generalizability to distribution shifts. We rigorously evaluate our method on 30 UEA multivariate and 85 UCR univariate datasets. Our empirical results demonstrate the superiority of the proposed method over 14 recent state-of-the-art methods.

---

[1]Clemson University, USA. [2]Arizona State University, USA. [3]Washington University in St. Louis, USA. [4]University of Arizona, USA [5]University of Massachusetts Boston, USA . Correspondence to: Xiwen Chen <xiwenc@g.clemson.edu>, Abolfazl Razi <arazi@clemson.edu>.

*Proceedings of the $42^{nd}$ International Conference on Machine Learning*, Vancouver, Canada. PMLR 267, 2025. Copyright 2025 by the author(s).

## 1. Introduction

Time series analysis is crucial in a wide range of applications, such as network traffic management (Ferreira et al., 2023), healthcare (Vrba & Robinson, 2001; Niknazar & Mednick, 2024), environmental science (Wu et al., 2023b), and economics (Sezer et al., 2020). This is due to the fact that much of the data they generate or collect naturally follows a time series format. Representative examples include stock prices in finance, electrocardiogram (ECG) signals in healthcare monitoring, and network traffic, activity logs, and social media data. In this work, we focus on the study of the time series classification (TSC) problem, one of the most important tasks in time series analysis (Bagnall et al., 2017; Chen et al., 2024; Ruiz et al., 2021; Wang et al., 2024c), which enables the categorization of these data sequences into distinct behaviors or outcomes.

There is broad consensus that time series data often experiences training/testing domain shifts, where the testing time series distribution differs significantly from that of the training data, resulting in a significant drop in performance (Kim et al., 2022b). This domain shift issue is largely attributed to several factors, including sensor variability, environmental changes, differences in measurement or preprocessing methods, and data collection at different times. A common approaches to mitigate domain shifts involve designing normalization to reduce domain-specific biases, enhance feature invariance, mitigate covariate shifts, and promote more generalizable representation learning (Ioffe & Szegedy, 2015; Ulyanov et al., 2016; Wu & He, 2018; Li et al., 2018). One representative normalization method is Reversible Instance Normalization (RevIN, Kim et al., 2022b; Liu et al., 2023), which has been widely used in time series forecasting (regression) tasks. However, adjusting normalization may pose significant challenges to model training, e.g., gradient instability, increased sensitivity to hyperparameters, difficulty in optimizing the loss landscape, and potential overfitting to the training domain. Consequently, this leads to a mismatch between training and inference loss (Dinh et al., 2017b; Zhou et al., 2021). More importantly, its effectiveness in time series classification remains **unexplored** in previous literature.

To fill this gap, our study starts the investigation of domain shifts in time series classification task across several datasets. We confirm that the domain shift issue is also common in time series classification tasks. We further conduct an additional analysis of `RevIN` in time series classification tasks, revealing that it is surprisingly ineffective this task. This motivates us to develop a method that is more tailored to domain shift issues in time series classification tasks. To this end, we propose a novel learning method based on Fisher information, which measures the amount of information an observable random variable carries about an unknown parameter (Ly et al., 2017). In this context, we can utilize it as a measure of a neural network's sensitivity to small changes in input data. However, directly applying Fisher information as a regularization term in gradient-based optimization suffers from a high computational and memory cost: (i) the Fisher Information Matrix (FIM), a matrix that contains the Fisher information for all parameters, is prohibitively large due to its quadratic complexity w.r.t. the number of parameters; (ii) it needs to back-propagate twice (the first pass for computing the FIM and the second for updating the neural network), which introduces relatively substantial computation. To this end, we utilize diagonal approximation and propose gradient re-normalization based on FIM to tackle these challenges.

We find our method is surprisingly effective and has a nice theoretical interpretation. First, it is easy to be integrated into the existing learning framework with automatic differentiation without an additional back-propagation. It only requires a $\mathcal{O}(n)$ memory complexity w.r.t. $n$ number of neural network parameters. Second, our method can guide the neural network to converge to a flat local minimum, potentially resulting in better generalizability in dealing with the issue of domain shifts (see e.g., Keskar et al., 2016; Zhang & Xu, 2024). Finally, we show that despite the constraints, the theoretical convergence rate remains on par with that of standard neural network training.

In summary, our contributions are two-fold: **(i)** We analyze the domain shift problem in time series classification and highlight the ineffectiveness of previous methods in addressing this issue in classification tasks. **(ii)** To resolve this, we propose *FIC-TSC*, a time series classification model trained with a novel Fisher Information Constraint. We rigorously evaluate our approach on 30 UCR multivariate time series classification datasets (Bagnall et al., 2018) and 85 UEA univariate time series classification datasets (Chen et al., 2015), demonstrating the superiority of our method over state-of-the-art methods.

## 2. Related Work

**Time Series Classification.** Traditional methods for TSC focus on similarity measurement techniques (Berndt & Clif-

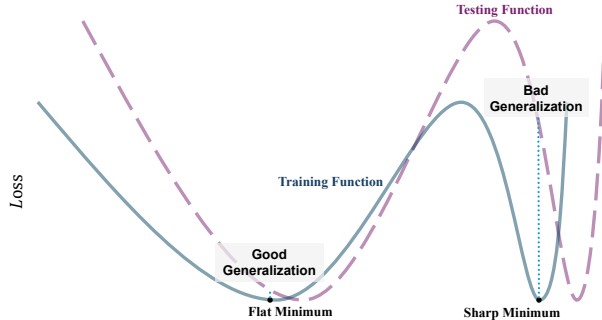

*Figure 1.* A conceptual visualization of Flat and Sharp Minima. The Y-axis indicates loss, and the X-axis represents the variables (neural network parameters). Under train (blue) and test (purple) data domains, due to potential distribution shift, the landscapes differ, i.e., with the same network parameter, the loss is often different. A flat minimum can potentially lead to a low test error, while a sharp minimum potentially leads to a high test error.

ford, 1994; Seto et al., 2015), while further, the advent of deep learning has transformed TSC by enabling automated feature extraction and improving performance significantly. They employ or develop based on different architectures, such as CNN/LSTM hybrid (Karim et al., 2019; Zhang et al., 2020), purely CNN (Ismail Fawaz et al., 2020; Li et al., 2021; Wu et al., 2023a), and Transformer (Zerveas et al., 2021; Nie et al., 2023; Foumani et al., 2024; Eldele et al., 2024). Instead of supervised learning, there are also recent methods that benefit from self-supervised pre-training, such as (Lin et al., 2023; Li et al., 2024). However, one significant issue is the non-convex nature of the optimization problem inherent in training deep neural networks. Due to the complex structure of these models, the loss landscape is often filled with numerous local minima. This non-convexity poses challenges in finding a global minimum or even a sufficiently good local minimum. Hence, our proposed method can be one of the elegant solutions to mitigate this issue and guide the neural network to converge to a flat minimum with desirable generalizability (for details, see Sec. 4.2).

**Sharpness and Model Generalizability.** In the context of deep learning optimization, sharpness refers to the curvature of the loss landscape. Several works have illustrated that sharper minima often lead to poor generalization performance (see e.g., Keskar et al., 2016; Neyshabur et al., 2017; Zhang & Xu, 2024). This is because models that converge to sharp minima may overfit the training data, leading to a larger generalization gap. In contrast, a flat minimum is less sensitive to the small perturbation of parameters, and hence, is more robust to domain shifts (see Fig. 1 for illustration). Accordingly, some works (Foret et al., 2020; Andriushchenko & Flammarion, 2022; Kim et al., 2022a; Yun & Yang, 2024) have been proposed to ac-

count for the sharpness of minima during training explicitly. They achieve this by modifying the traditional gradient-based optimization process: at each iteration, they compute the loss gradient w.r.t. parameters perturbed with small noise. This perturbation encourages the optimizer to move toward regions of the loss landscape where the loss remains low over a larger neighborhood (i.e., flatter minima). Recent work (Ilbert et al., 2024) designed for forecasting also falls into this category. However, the obvious issue is that they need to back-propagate twice at each training iteration, which introduces considerable computational and memory overheads. Consequently, their methods may have not been widely applied. In contrast, our method only requires a single back-propagation at each iteration, just like the standard model training. More importantly, we demonstrate that despite the inclusion of the constraint, the convergence rate of our method is on par with standard training processes.

**Fisher Information in Time Series Analysis.** We notice some conventional approaches (i.e., not based on deep learning) have utilized Fisher information as a tool for time series analysis. For example, authors in (Dobos & Abonyi, 2013) use the Fisher Information to identify changes in the statistical properties of time-series data, facilitating the segmentation of the data into homogeneous intervals. Likewise, authors in (Telesca & Lovallo, 2017; Wang & Shang, 2018; Contreras-Reyes & Kharazmi, 2023) employ Fisher information and Shannon Entropy to measure the temporal properties of time series in dynamic systems. In this context, the parameters often represent statistical properties of the series, like mean, variance, or even parameters defining windowed segments of the series for localization analysis. Therefore, our methods are essentially different from theirs, where we propose the Fisher information-based constraint to guide the optimization of the neural networks.

**Domain adaptation.** It addresses distribution shifts by enabling models trained on a source domain to generalize to a related target domain. Common strategies include aligning feature distributions by minimizing statistical distances (Chen et al., 2020) or using adversarial training to make features indistinguishable across source and target domains (Purushotham et al., 2017; Jin et al., 2022). In contrast, our method is orthogonal to these approaches. It does not require access to target domain data or labels during training, making it suitable when the target distribution is unknown or unavailable. Importantly, domain adaptation could be applied as a post-training or downstream enhancement once target domain data becomes available. Hence, our method and these techniques can be complementary.

## 3. Preliminary Analysis

**Distribution Discrepancy.** First, we validate our conjecture of the distribution discrepancy between the training and

testing sets on several datasets. For convenience, following (Kim et al., 2022b), in each dataset, we illustrate the histogram of the first channels of all samples from both training and test sets, offering a statistical perspective to interpret the distribution. As shown in Fig. 2, the observation is aligned with (Kim et al., 2022b), and it is evident that the distribution discrepancy is a common phenomenon across different datasets from two perspectives: (i) distribution between the entire training and testing sets and (ii) distributions of the same class from the training set and test set. We further employ the Wasserstein-1 distance to evaluate the distribution distance, which can be mathematically defined as below,

**Definition 1.** *((Peyré et al., 2019)) The **Wasserstein-1 distance** between two probability distributions $P$ and $Q$ with cumulative distribution functions (CDFs) $F_P(x)$ and $F_Q(x)$ is defined as:*

$$W_1(P, Q) = \inf_{\gamma \in \Gamma(P,Q)} \mathbb{E}_{(x,y) \sim \gamma}[d(x, y)] \quad (1)$$

*where $\Gamma(P, Q)$ is the set of all couplings (joint distributions) $\gamma(x, y)$ with marginals $P$ and $Q$, and $d(x, y) = |x - y|$ is a absolute value distance function between points $x$ and $y$.*

Here, we used the discrete case Wasserstein-1 distance to calculate the distance between 1D discrete distributions (Ramdas et al., 2017). The dissimilarity matrices of these datasets are also shown in Fig. 2, where each element denotes class distribution from different sets. For example, the upper left element represents the distance between class 1 from the training set and class 1 from the test set. It is worth mentioning that we apply min-max normalization here for better visualization. We observe that the within-class distance (i.e., the same class distributions from different sets) can even be equal or greater than between-class distance (i.e., different classes from the same set), such as `FaceDetection` and `SelfRegulationSCP1`. These results reveal some potential reasons why the general classification methods have poor performance on time series data, and they motivate us to develop a method that can be somewhat robust for the distribution discrepancy.

**Effectiveness of `RevIN`.** The invention of Reversible Instance Normalization (Kim et al., 2022b) is designed to solve the train/test domain shift and has substantially promoted the performance of machine learning methods for sequence-to-sequence regression tasks, particularly forecasting tasks. This method applies instance normalization at the beginning to remove the non-stationary information (i.e., subtracting the mean and dividing by the standard deviation. This leads the sequence to be a zero-mean and standard deviation of one) and perform denormalization at

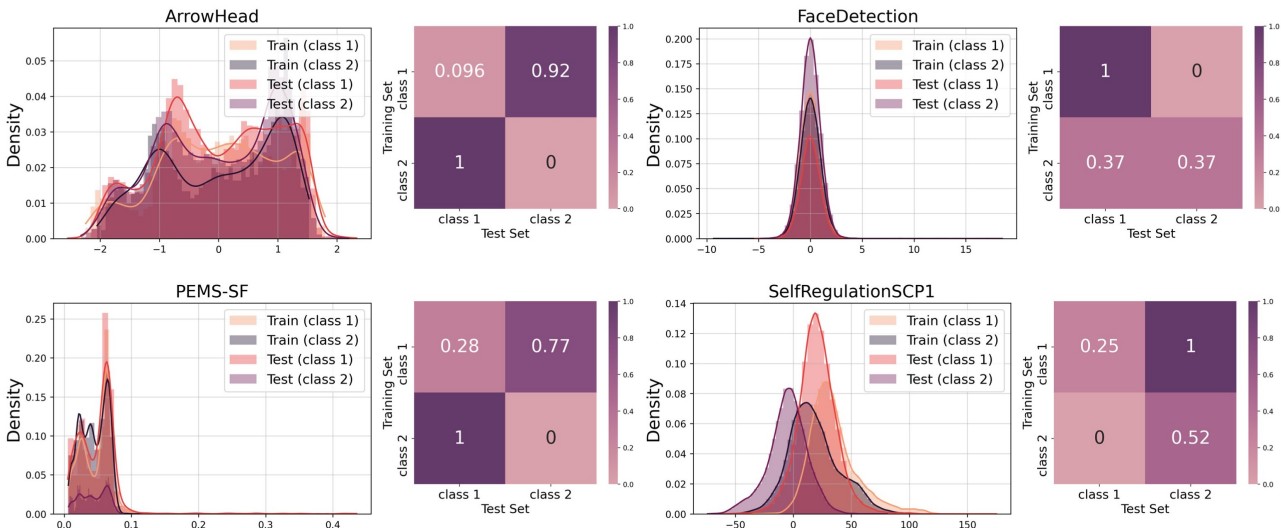

*Figure 2.* Histograms representing sequences from the two selected classes of exemplary datasets with train/test distribution shift. The dissimilarity matrix illustrates the min-max normalized `Wasserstein-1 distance` between class distribution from different sets. A lower value implies two distributions are more similar. It is observed that distribution shifts exist between the entire training and testing sets and within the same classes across these sets.

the end to restore that information. In the classification task, only normalization at the beginning is appropriate since the output is logits rather than a sequence. Hence, we just use the term `IN` in the following text. However, this technique has not illustrated any benefits in classification. This is reasonable since although `IN` reduces the distribution shifts between the entire training and testing sets, it can also reduce the distances between different class distributions within the same set. We show this fact in Fig. 3. We further empirically evaluate this technique on ten datasets: **EC:** EthanolConcentration, **FD:** FaceDetection, **HW:** Handwriting, **HB:** Heartbeat, **JV:** JapaneseVowels, **SCP1:** SelfRegulationSCP1, **SCP2:** SelfRegulationSCP2, **SAD:** SpokenArabicDigits, **UW:** UWaveGestureLibrary, and **PS:** PEMS-SF. According to the results shown in Fig. 4, `IN` does not positively affect most datasets' performance. In some datasets, `IN` even suppresses the performance. Applying `IN` finally lowers the average accuracy, aligning with our conjecture. Therefore, `IN` is not a desirable solution to solve the distribution discrepancy issue for the classification task. Thus, we seek to propose a new method for addressing this issue, and we will discuss it in the next section.

## 4. Method

### 4.1. Problem Formulation

Time series data can be presented as $\{\boldsymbol{X}_1, \cdots, \boldsymbol{X}_n\}$, where $\boldsymbol{X}_i = \{\boldsymbol{x}_i^1, \cdots, \boldsymbol{x}_i^T\}$ denotes a time sequences containing $T$ time points, with each time point $\boldsymbol{x}_i^\top \in \mathbb{R}^d$ being a vector with a dimension of $d$. The goal of the task is to learn a machine-learning model that directly maps feature space $\mathcal{X}$ to target space $\mathcal{Y}$. The $d = 1$ implies this sequence is

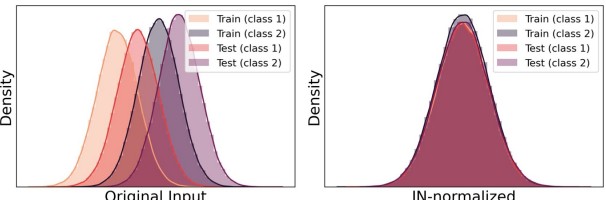

*Figure 3.* An illustration of the negative effect of Instance Normalization (IN). **Left**: The original input, and **Right**: Input after applying `IN`. It is observed that `IN` can reduce the difference of two class distributions. This may be disadvantageous for classification.

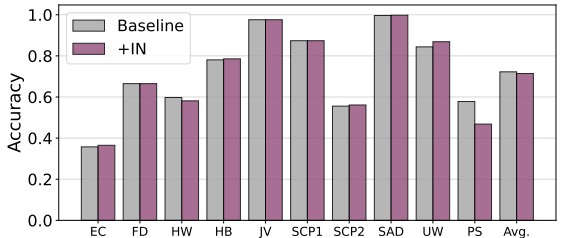

*Figure 4.* Comparison of classification accuracy between baseline and the model applying `IN`. **Avg.** indicates Average accuracy across all datasets. It is observed that `IN` does *not have any positive effect* on the model for most datasets. A statistical test is conducted in Appendix E.1.

univariate, while $d > 1$ is multi-variate. Our work can be generalized for both cases.

### 4.2. Learning with the FIM constraint

As discussed in Section 3, while normalization can mitigate train-test distribution shifts, it may also reduce inter-

class distances, potentially hindering training effectiveness. Hence, we believe this is not a feasible solution for the classification task. In contrast, we expect to train a more robust network to handle the distribution difference among training and test sets from a gradient optimization perspective. To this end, we propose an FIC-constrained strategy to guide the optimization of the network.

FIM is used to measure the amount of information/sensitivity about unknown parameters $\Theta$ carried by the data $\mathcal{D}$, and lower Fisher information often represents the parameters that are less sensitive to a small change of the data, which potentially improves its robustness. We first give its definition here:

**Definition 2.** *((Kay, 1993)) Given a model parameterized by $\Theta$ and an observable random variable $\mathcal{D}$, the Fisher Information Matrix (FIM) is defined as,*

$$\mathcal{F}(\Theta) \tag{2}$$
$$=\mathbb{E}_{p(\mathcal{D}|\Theta)}\left[(\nabla_\Theta \log p(\mathcal{D} \mid \Theta))(\nabla_\Theta \log p(\mathcal{D} \mid \Theta))^\top\right],$$

*where $\log p(\mathcal{D} \mid \Theta)$ denotes the log-likelihood.*

*Remark* 1. In the context of the classification task, this log-likelihood can be interpreted as the negative cross-entropy (maximizing the log-likelihood is empirically equivalent to minimizing cross-entropy loss). To this end, we present the commonly used cross-entropy loss here.

$$\mathcal{L}(\{\mathcal{D}_i\}_{i=1}^n \mid \Theta) = -\frac{1}{n}\sum_{i=1}^n \log \hat{P}(\mathcal{D}_i \mid \Theta), \tag{3}$$

where $\mathcal{D}_i$ denotes a training pair, $n$ denotes the number of samples, $\log \hat{P}(\mathcal{D}_i \mid \Theta)$ denotes the logarithm of the predicted probability of the true label.

*Remark* 2. Directly computing FIM suffers from large computation and memory cost $\mathcal{O}(n^2)$, where $n$ denotes the number of parameters.

Remark 2 naturally motivates the simplification of the computation to facilitate the broader adoption of our method, particularly for users with limited computational resources. Hence, we apply the *diagonal approximation*, which ignores the off-diagonal elements and only interest $\text{diag}(\mathcal{F}(\Theta))$ in the FIM. Here, $\text{diag}(\cdot)$ denotes the diagonal elements of a matrix. This drastically reduces the computational burden and memory needs to $\mathcal{O}(n)$. More importantly, it still has reasonable information, and we will employ this approximation in our theoretical justification later. Empirically, given a set of data pairs $\{\mathcal{D}_i\}_{i=1}^n$, the FIM is estimated as,

$$\text{diag}(\mathcal{F}(\Theta)) = \nabla_\Theta \mathcal{L}(\Theta) \circ \nabla_\Theta \mathcal{L}(\Theta), \tag{4}$$

where $\circ$ denotes the element-wise product and again the off-diagonal elements of $\mathcal{F}(\Theta)$ is zero.

Additionally, simply applying FIM as regularization suffers double backward passes, where the first backward is used to compute the FIM and the second backward is used to impose regularization for updating the neural network. This again will introduce extra computation. Therefore, to avoid the computational cost of performing double backward passes, we constrain the optimization by introducing a normalization strategy. Let $\epsilon$ be the pre-defined upper-bound and $\|\mathcal{F}\|_1$ denote the *entrywise 1-norm*[1] of the FIM. If $\|\mathcal{F}\|_1 \geq \epsilon$, the gradient of each parameter $\theta_i$ is normalized as follows:

$$\nabla_\Theta \mathcal{L}(\Theta) \leftarrow \sqrt{\frac{\epsilon}{\|\mathcal{F}\|_1}} \nabla_\Theta \mathcal{L}(\Theta). \tag{5}$$

It is worth mentioning that under the diagonal approximation, $\|\mathcal{F}(\Theta)\|_1 = \|\text{diag}(\mathcal{F}(\Theta))\|_1$, where $\text{diag}(\mathcal{F}(\Theta))$ is a vector.

Considering these two aspects mentioned above, we summarize the proposed *FIC-constrained optimization* as,

$$\min_\Theta \mathcal{L}(\mathcal{D}; \Theta) \quad s.t. \|\mathcal{F}(\Theta)\|_1 \leq \epsilon. \tag{6}$$

The complete algorithmic summary is provided in Algorithm 1, located in Appendix B.

**Theoretical Justification.** Now, we will delve into the theoretical support of our method and illustrate our method has the potential to lead to a better convergence with a theoretically guaranteed rate. We first note that there is a strong relationship between FIM and the second derivative of the loss,

**Lemma 1.** *At a local minimum, the expected Hessian matrix of the negative log-likelihood is asymptotically equivalent to the Fisher Information Matrix, w.r.t $\Theta$, which is presented as,*

$$\mathbb{E}_{p(\mathcal{D}|\Theta)}\left[\nabla^2(-\log p(\mathcal{D} \mid \Theta))\right] = \mathcal{F}. \tag{7}$$

*Remark* 3. See Appendix A for the proof. As previously discussed, minimizing the negative log-likelihood is equivalent to minimizing the loss.

Accordingly, we realize that FIM can be bridged to the principle of *sharpness*. Sharpness measures the curvature around a local minimum in a neural network's loss landscape. It is formally defined as follows,

**Definition 3.** *(Keskar et al., 2016) For a non-negative valued loss function $\mathcal{L}_\Theta$, given $\mathcal{B}_2(\alpha, \Theta)$, a Euclidean ball with radius $\alpha$ centered at $\Theta$, we can define the*

---

[1]this norm will be utilized in the subsequent operations involving matrices

*α-sharpness as,*

$$\alpha\text{-sharpness} \propto \frac{\max_{\Theta' \in \mathcal{B}_2(\alpha,\Theta)} \mathcal{L}_{\Theta'} - \mathcal{L}_\Theta}{1 + \mathcal{L}_\Theta}. \quad (8)$$

We observe this can be efficiently estimated at a local minimum,

**Corollary 1.** *At a local minimum, the upper bound of α-sharpness is able to be approximated via Taylor expansion as*

$$\alpha\text{-sharpness} \propto \frac{\alpha^2 \|\mathcal{F}\|_1}{2(1 + \mathcal{L}(\Theta))}. \quad (9)$$

*Proof.* Please refer to Appendix A. □

It is worth reiterating that a lower sharpness often implies higher generalizability (Keskar et al., 2016; Zhang & Xu, 2024). Hence, according to Corollary 1, we can conclude:

**Proposition 1.** *(Achievability.) With the same training loss, appropriately constraining the Fisher information can potentially converge to flatter minima and result in better generalizability.*

Now, we are interested in the convergence of the proposed method. We offer the analysis under some mild and common assumptions.

**Theorem 4.** *Consider a L-Lipschitz objective function $\mathcal{L}(\Theta)$, defined such that for any two points $\Theta_1$ and $\Theta_2$, the inequality $\|\nabla\mathcal{L}(\Theta_1) - \nabla\mathcal{L}(\Theta_2)\| \leq L\|\Theta_1 - \Theta_2\|$ holds. By appropriately choosing the learning rate $\eta$ and the FIC constraint $\epsilon$, the convergence rate of gradient descent can be expressed as $\mathcal{O}(\frac{1}{T})$, where $T$ represents the number of iterations.*

*Proof.* Please refer to Appendix A. □

*Remark 5.* This suggests the empirical convergence time may be a little slow due to the constraint, while asymptotically, the order of the theoretical convergence rate is not explicitly related to the constraint w.r.t $T$.

## 5. Experiment

**Setup.** To thoroughly evaluate our method, we conducted experiments on both MTSC and UTSC tasks. Specifically, we utilized the UEA multivariate datasets (30 datasets) (Bagnall et al., 2018) and the UCR univariate datasets (85 datasets) (Chen et al., 2015), which are among the most comprehensive collections in the field. These datasets involve a wide range of applications, including but not limited to

traffic management, human activity recognition, sensor data interpretation, healthcare, and complex system monitoring. A summary of the datasets is shown in Table 1. Please refer to Appendix C for more details about the datasets. The data pre-processing follows (Wu et al., 2023a; Foumani et al., 2024).

*Table 1.* A summary of UEA and UCR datasets.

| Dataset | Statistic | # of Variates | Length | Training Size | Test Size | # of classes |
|---------|-----------|---------------|--------|---------------|-----------|--------------|
| UEA 30 | min | 2 | 8 | 12 | 15 | 2 |
| | max | 1345 | 17984 | 30000 | 20000 | 39 |
| UCR 85 | min | 1 | 24 | 16 | 20 | 2 |
| | max | 1 | 2709 | 8926 | 8236 | 60 |

**Baselines.** We compare our method with multiple alternative methods, including recent advanced representation learning-based approaches: TsLaNet (Eldele et al., 2024), GPT4TS (Zhou et al., 2023), TimesNet (Wu et al., 2023a), PatchTST (Nie et al., 2023), Crossformer (Zhang & Yan, 2023), TS-TCC (Eldele et al., 2021), and including classification-specific approaches: ROCKET (Dempster et al., 2020), InceptionTime (ITime) (Ismail Fawaz et al., 2020), ConvTran (Foumani et al., 2024) (MTSC only), MILLET (Early et al., 2024), TodyNet (Liu et al., 2024) (MTSC only), HC2 (Middlehurst et al., 2021) (UTSC only), and Hydra-MR (Dempster et al., 2023) (UTSC only). Finally, we involve the model with a single linear layer for comparison.

**Implementation.** As a proof-of-concept, we implement our algorithm on a network based on ITime (Ismail Fawaz et al., 2020), while we modify the original one linear classier to a two-layer MLP. We apply two hyperparameter strategies for our implementation: (i) **Uni.**: For the sake of the robustness and rigor of our method, we apply universal hyperparameters for all datasets in this model. We fixed the $\epsilon$ to 2 and mini-batch size to 64 for UEA datasets and 16 for UCR datasets, and (ii) **Full**: To fully explore the ability of our method, we perform a grid search for hyperparameters for each dataset. Specifically, we search the mini-batch size from $\{16, 32, 64, 128\}$ and $\epsilon$ from $\{2, 4, 10, 20\}$. We place their performance aside from the main results. Please refer to Appendix D for more details about the implementation.

**Main Results.** We report results on UEA multivariate datasets in Table 2 averaged over 5 runs. It is worth mentioning that this result only includes 26 datasets since several methods (following TsLaNet (Eldele et al., 2024)) are not able to process the remaining datasets due to memory or computational issues. However, our method does not have any obstacles to learning on these datasets, and hence, we report the comprehensive results with additional metrics (i.e., balanced precision, F1 score, precision, recall) in Table 4 and Appendix F. It is known that adjusting hyperparameters is crucial for maintaining optimal performance across diverse datasets, and we note that this strategy is employed by

*Table 2.* Comparison of classification accuracy by our method with recent alternative methods for multivariate time series classification (26 UEA datasets). We highlight the best results in **bold** and the second-best results with underlining. *Uni.* indicates we set the consistent hyperparameters for all datasets, while *Full* indicates we perform dataset-wise grid search for hyperparameters. To ensure the robustness and rigor of our approach, we included *Full* at the end but did not participate in the comparison. The accuracy is averaged over 5 runs.

| Dataset | Ours (Uni.) | MILLET ICLR'24 | TodyNet Inf. Sci.'24 | ConvTran ECML'23 | ROCKET ECML'20 | ITime DMKD'20 | TsLaNet ICML'24 | GPT4TS NeurIPS'23 | TimesNet ICLR'23 | CrossFormer ICLR'23 | PatchTST ICLR'23 | TS-TCC IJCAI'21 | Linear | Ours (Full) |
|---|---|---|---|---|---|---|---|---|---|---|---|---|---|---|
| ArticularyWordRecognition | **99.3** | 99.0 | 98.2 | 98.3 | **99.3** | 98.5 | 99.0 | 93.3 | 96.2 | 98.0 | 97.7 | 98.0 | 93.1 | 99.8 |
| AtrialFibrillation | **56.7** | 43.3 | 46.7 | 40.0 | 20.0 | 44.0 | 40.0 | 33.3 | 33.3 | 46.7 | 53.3 | 33.3 | 46.7 | 60.0 |
| BasicMotions | **100.0** | **100.0** | **100.0** | **100.0** | **100.0** | **100.0** | **100.0** | 92.5 | **100.0** | 90.0 | 92.5 | **100.0** | 85.0 | 100.0 |
| Cricket | **100.0** | **100.0** | 98.6 | **100.0** | 98.6 | 98.6 | 98.6 | 8.3 | 87.5 | 84.7 | 84.7 | 93.1 | 91.7 | 100.0 |
| Epilepsy | **98.9** | 98.6 | 96.4 | 98.6 | 98.6 | 97.2 | 98.6 | 85.5 | 78.1 | 73.2 | 65.9 | 97.1 | 60.1 | 100.0 |
| EthanolConcentration | 39.2 | 32.3 | 42.4 | 36.1 | **42.6** | 36.3 | 30.4 | 25.5 | 27.7 | 35.0 | 28.9 | 32.3 | 33.5 | 40.1 |
| FaceDetection | 68.4 | **69.2** | 67.7 | 67.2 | 64.7 | 66.6 | 66.8 | 65.6 | 67.5 | 66.2 | 69.0 | 63.1 | 67.4 | 69.2 |
| FingerMovements | **65.0** | 64.0 | 62.5 | 56.0 | 61.0 | 60.0 | 61.0 | 57.0 | 59.4 | 64.0 | 62.0 | 44.0 | 64.0 | 71.5 |
| HandMovementDirection | 49.3 | 50.7 | 54.1 | 40.5 | 50.0 | 44.9 | 52.7 | 18.9 | 50.0 | 58.1 | 58.1 | **64.9** | 58.1 | 54.7 |
| Handwriting | 61.6 | **69.8** | 47.9 | 37.5 | 48.5 | 60.1 | 57.9 | 3.8 | 26.2 | 26.2 | 26.0 | 47.8 | 22.5 | 73.0 |
| Heartbeat | **81.0** | 76.8 | 79.0 | 78.5 | 69.8 | 78.8 | 77.6 | 36.6 | 74.5 | 76.6 | 76.6 | 77.1 | 73.2 | 82.0 |
| InsectWingbeat | 71.1 | **72.0** | 63.0 | 71.3 | 41.8 | 70.9 | 10.0 | 10.0 | 10.0 | 10.0 | 10.0 | 10.0 | 10.0 | 72.3 |
| Japanese Vowels | 99.1 | **99.6** | 97.6 | 98.9 | 95.7 | 95.7 | 99.2 | 98.1 | 97.8 | 98.9 | 98.7 | 97.3 | 97.8 | 99.6 |
| Libras | 79.4 | 91.1 | 84.4 | 92.8 | 83.9 | 68.1 | **92.8** | 79.4 | 77.8 | 76.1 | 81.1 | 86.7 | 73.3 | 91.7 |
| LSST | 65.3 | 50.2 | 65.1 | 61.6 | 54.1 | 52.9 | **66.3** | 46.4 | 59.2 | 42.8 | 67.8 | 49.2 | 35.8 | 66.8 |
| MotorImagery | **65.0** | 61.5 | 64.0 | 56.0 | 53.0 | 60.2 | 62.0 | 50.0 | 51.0 | 61.0 | 61.0 | 47.0 | 61.0 | 68.5 |
| NATOPS | **98.9** | 98.3 | 95.6 | 94.4 | 83.3 | 96.8 | 95.6 | 91.7 | 81.8 | 88.3 | 96.7 | 96.1 | 93.9 | 99.2 |
| PEMS-SF | 79.2 | 63.0 | **96.1** | 82.8 | 75.1 | 60.4 | 83.8 | 87.3 | 88.1 | 82.1 | 88.4 | 86.7 | 82.1 | 87.9 |
| PenDigits | 97.6 | 92.6 | 98.7 | 98.7 | 97.3 | 95.7 | **98.9** | 97.7 | 98.2 | 93.7 | 99.2 | 98.5 | 92.9 | 98.5 |
| PhonemeSpectra | 31.3 | **34.3** | 31.2 | 30.6 | 17.6 | 30.1 | 17.8 | 3.0 | 18.2 | 7.6 | 11.7 | 25.9 | 7.1 | 32.3 |
| RacketSports | 89.8 | 88.2 | 84.5 | 86.2 | 86.2 | 87.9 | **90.8** | 77.0 | 82.6 | 81.6 | 84.2 | 84.9 | 79.0 | 90.8 |
| SelfRegulationSCP1 | 90.1 | 88.1 | 90.4 | 91.8 | 84.6 | 88.1 | 91.8 | 91.5 | 77.4 | **92.5** | 89.8 | 91.1 | 88.4 | 90.3 |
| SelfRegulationSCP2 | 59.4 | 59.4 | 59.4 | 58.3 | 54.4 | 56.4 | **61.7** | 51.7 | 52.8 | 53.3 | 54.4 | 53.9 | 51.7 | 60.8 |
| SpokenArabicDigits | **99.9** | **99.9** | 99.1 | 99.5 | 99.2 | 99.7 | **99.9** | 99.4 | 98.4 | 96.4 | 99.7 | 99.8 | 96.7 | 100.0 |
| StandWalkJump | **63.3** | 53.3 | 36.7 | 33.3 | 46.7 | 49.3 | 46.7 | 33.3 | 53.3 | 53.3 | 60.0 | 40.0 | 60.0 | 66.7 |
| UWaveGestureLibrary | 90.2 | 90.8 | 86.3 | 89.1 | **94.4** | 84.8 | 91.3 | 84.4 | 83.1 | 81.6 | 80.0 | 86.3 | 81.9 | 93.0 |
| **Avg.** | **76.9** | 74.8 | 74.8 | 73.0 | 70.0 | 72.5 | 72.7 | 58.5 | 66.6 | 66.8 | 69.1 | 69.4 | 65.6 | 79.6 |

*Table 3.* Comparison of classification accuracy by our method with recent alternative methods for univariate time series classification (85 UCR datasets). We highlight the best results in **bold** and the second-best results with underlining. Please see the full results in Appendix F.

| Method | Ours (Uni.) | MILLET | HC2 | Hydra-MR | ITime | ROCKET | TsLaNet | GPT4TS | TimesNet | CrossFormer | PatchTST | TS-TCC | Linear | Ours (Full) |
|---|---|---|---|---|---|---|---|---|---|---|---|---|---|---|
| AVg. | **86.2** | 85.6 | 86.0 | 85.7 | 85.6 | 83.1 | 83.2 | 61.6 | 65.3 | 73.4 | 71.8 | 75.1 | 69.7 | 87.3 |

*Table 4.* Classification results obtained using our method, evaluated across a comprehensive set of metrics. *Acc.*: Accuracy, *Bal. Acc.*: Balanced Accuracy, *F1*: F1 score, *P*: Precision, *R*: Recall.

| Dataset | Model | Acc. | Bal. Acc. | F1 | P | R |
|---|---|---|---|---|---|---|
| UEA 30 | Uni. | 78.0 | 76.7 | 76.3 | 78.0 | 76.7 |
| UEA 30 | Full | 80.9 | 79.7 | 79.3 | 80.7 | 79.7 |
| UCR 85 | Uni. | 86.2 | 83.4 | 83.3 | 85.2 | 83.4 |
| UCR 85 | Full | 87.3 | 84.8 | 84.8 | 86.4 | 84.8 |

| Comparison | p-Value |
|---|---|
| **Uni.** vs **MILLET** | 0.039 |
| **Uni.** vs **TodyNet** | 0.020 |

A similar conclusion can be reached on UCR univariate datasets as the results presented in Table 3, where we achieve a 0.2% gain over the previous SOTA method (HC2). Our full model can obtain a 1% further enhancement over our method with universal hyperparameters. This is reasonably significant as we evaluate models across 85 datasets as optimal hyperparameters in different datasets can be various. We also realize that HC2 is an ensemble method of multiple different classifiers with expensive computational costs. Therefore, our method not only improves performance but also offers a more efficient solution. In conclusion, all these results highlight the effectiveness of our method.

several models, such as TodyNet and TimesNet. However, the main observation is that even with universal hyperparameters, our method can achieve the best overall performance and obtain a gain over previous SOTA methods (TodyNet and MILLET) with an average of 2.1% accuracy improvement.

We also present the single-tailed Wilcoxon signed rank test to compare our methods against the most competitive candidates (**MILLET** and **TodyNet**) in the following table, which confirms the statistical superiority. More surprisingly, our full model can achieve an average accuracy of over 79.6%, with an additional 2.7% boosting of our model with universal hyperparameters.

## 6. Model Analysis

In this section, we will delve into the analysis of our model from several perspectives. We will use the ten datasets as the same as those used in Section 3 since they are commonly selected by previous works (Zerveas et al., 2021;

Wu et al., 2023a). We use the model with the universal hyperparameters in the following investigations.

**Ablation Analysis.** We first investigate the effectiveness of our methods over baseline in Fig. 5. It is evident that training with our proposed constraint can obtain accuracy gain in all datasets. Particularly, we observe that even though there are fluctuations on some datasets, setting $\epsilon$ to 2 can obtain a relatively considerable gain with a 4% average improvement. We include statistical test in Appendix E.2.

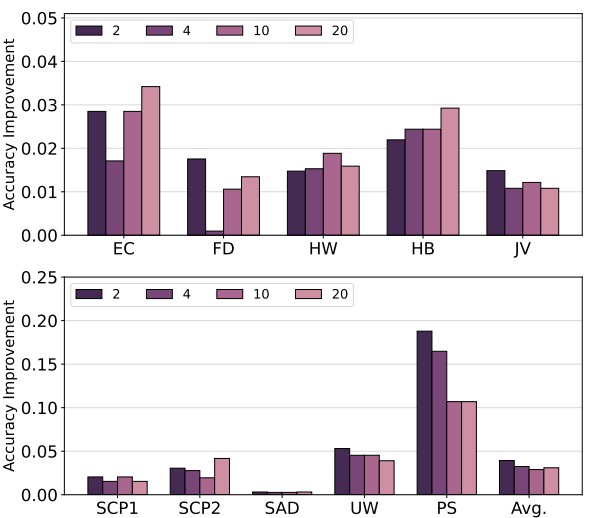

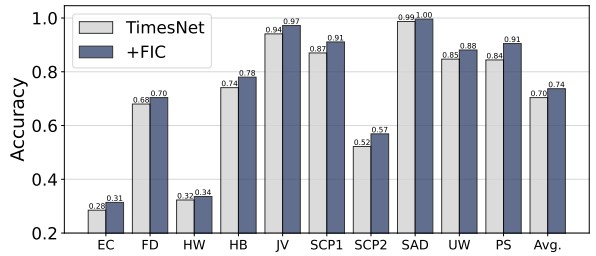

*Figure 5.* The accuracy improvement of our method over the baseline (i.e. standard training without using any constraint). We present the average accuracy improvement at the end.

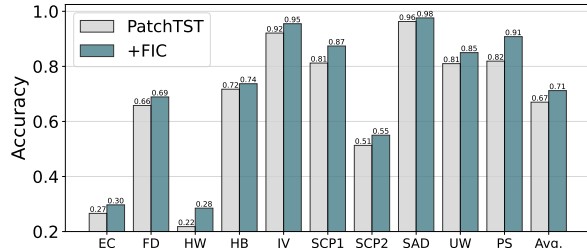

*Figure 7.* Comparison of the classification accuracy between original PatchTST and it after applying our proposed method.

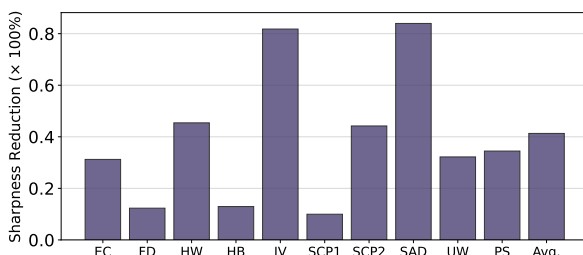

*Figure 8.* The **sharpness reduction** by applying our method.

landscape caused by our method. Specifically, we compute the sharpness (Eq. 9) between our and the baseline models after well training. We reiterate here that a lower sharpness often implies a better generalization. We show the results in Fig. 8, which confirms that using our method can obtain an average 40% reduction in the sharpness across all datasets. We illustrate this by visualizing the landscape in Fig. 9.

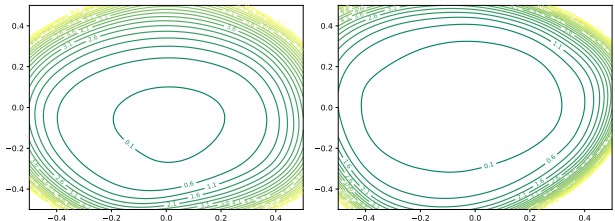

*Figure 9.* The landscape (error contour lines) of the well-trained network by **baseline (Left)** and **Ours (Right)**. The contour lines of our method are more spread out around the center compared to the baseline, indicating a flatter region.

*Figure 6.* Comparison of the classification accuracy between original TimesNet and it after applying our proposed method.

**Case Study on TimesNet and PatchTST.** We conduct a case study on TimesNet (Nie et al., 2023) and PatchTST (Wu et al., 2023a) to illustrate the proposed FIC can be generalized to other models. We implement it using the open source code[2] with default hyperparameters. The results shown in Figs. 6 and 7, where both models can obtain around 4% gain in accuracy. This confirms the effectiveness of our method.

**Landscape Analysis.** Here, we validate the change in the

**Comparison with SAM.** We also want to highlight the advantage of our method over previous sharpness-aware minimization (SAM) (Foret et al., 2020; Ilbert et al., 2024). For the sake of the fair test, we fix the mini-batch size to 64 for both cases. We evaluate the accuracy and the runtime as shown in Table 5. This result demonstrates that our method improves accuracy by 2.6% and enhances efficiency, reducing the computation time by half.

**Explicit Domain Shift Scenario.** Here, we adopt a realistic evaluation scenario. In real-world healthcare applications, datasets are commonly partitioned by patient, such that individuals included in the training set are distinct from those

---

[2]https://github.com/thuml/Time-Series-Library

*Table 5.* Comparison of classification accuracy and runtime per iteration by SAM and our method. We fixed both mini-batch sizes to 64 for a fair comparison. A statistical test is conducted in Appendix E.3.

| Metrics | Accuracy($\uparrow$) | | Runtime (s)($\downarrow$) | |
|---|---|---|---|---|
| Dataset | SAM | Ours | SAM | Ours |
| EC | 36.1 | 39.2 | 0.103 | 0.046 |
| FD | 61.0 | 68.4 | 0.057 | 0.036 |
| HW | 58.6 | 61.6 | 0.468 | 0.182 |
| HB | 79.0 | 81.0 | 0.050 | 0.032 |
| JV | 98.7 | 99.1 | 0.045 | 0.028 |
| SCP1 | 88.1 | 90.1 | 0.061 | 0.032 |
| SCP2 | 60.6 | 59.4 | 0.072 | 0.036 |
| SAD | 99.9 | 100.0 | 0.203 | 0.091 |
| UW | 89.7 | 90.2 | 0.048 | 0.028 |
| PS | 69.9 | 79.2 | 0.472 | 0.185 |
| Avg. | 74.2 | **76.8** | 0.158 | **0.070** |

*Table 6.* Comparison of classification accuracy on healthcare datasets with explicit domain shift. We include Medformer (Wang et al., 2024a), a recent SOTA method, for reference.

| Dataset | Method | Acc. | Bal. Acc. | F1 | P | R |
|---|---|---|---|---|---|---|
| TDBrain | Baseline | 93.0 | 93.0 | 93.0 | 93.4 | 93.0 |
| | +FIC | **96.2** | **96.2** | **96.2** | **96.3** | **96.2** |
| | Medformer | 89.6 | – | 89.6 | 89.7 | 89.6 |
| ADFTD | Baseline | 46.0 | 43.9 | 44.0 | 44.1 | 43.9 |
| | +FIC | 52.8 | 49.9 | 48.3 | **52.3** | 49.9 |
| | Medformer | **53.3** | – | **50.7** | 51.0 | **50.7** |
| PTB-XL | Baseline | 73.9 | 59.2 | 60.0 | 65.4 | 59.2 |
| | +FIC | **75.1** | **61.4** | **63.0** | **68.6** | **61.4** |
| | Medformer | 72.9 | – | 62.0 | 64.1 | 60.6 |
| SleepEDF | Baseline | 85.1 | 74.3 | 74.8 | 76.5 | 74.3 |
| | +FIC | **86.7** | **75.5** | **75.5** | **78.3** | **75.5** |
| | Medformer | 82.8 | – | 71.1 | 71.4 | 74.7 |

in the testing set. This setup introduces an explicit domain shift, as highlighted by Wang et al. (2024b), due to inter-patient differences in physiological characteristics, signal noise, and device-specific variations. We select four popular publicly available datasets: TDBrain (Van Dijk et al., 2022), ADFTD (Miltiadous et al., 2023b;a), PTB-XL (Wagner et al., 2020), and SleepEDF (Kemp et al., 2000). As shown in Table 6, our method can consistently obtain improvement over the baseline, and can outperform Medformer (Wang et al., 2024a), a recent SOTA method, in most cases.

**Discussion.** We have included additional discussion on model analysis. please refer to Appendix G.

## 7. Conclusion

In this work, we propose FIC-TS, a novel learning strategy that leverages Fisher information as a constraint to address train/test domain shifts in time series classification. Our approach is both theoretically sound and computationally efficient, with empirical results that strongly support our theoretical insights. Specifically, our method not only achieves superior performance on both univariate and multivariate time series datasets compared to several state-of-the-art methods, but also leads to a notable reduction in sharpness. These observations confirm that constraining Fisher information encourages flatter minima, ultimately enhancing generalization.

## Acknowledgment

This material is based upon the work supported by the National Science Foundation under Grant Number CNS-2204721. It is also supported by our collaborative project with MIT Lincoln Lab under Grant Numbers 2015887 and 7000612889. Additionally, this work was partially supported by National Institutes of Health, United States (R01EY032125 and R21AG065942) and the State of Arizona via the Arizona Alzheimer Consortium.

## Impact Statement

This paper proposes FIC-TSC, a novel training framework for time series classification that leverages Fisher information as a constraint. Our approach effectively mitigates the domain shift problem and enhances model generalization by guiding optimization toward flatter minima.

**(i) Theoretical View.** We establish a Fisher Information Constraint (FIC) to regulate training, theoretically linking it to sharpness reduction and generalization under distribution shifts. Our framework maintains competitive convergence rates while improving robustness.

**(ii) Empirical Validation.** Our empirical analysis strongly aligns with our theoretical findings, confirming that constraining Fisher information effectively reduces sharpness and enhances generalization.

**(iii) Applicability.** FIC-TSC is designed to be broadly applicable across diverse time series classification tasks. We validate its effectiveness on 30 UEA multivariate and 85 UCR univariate datasets, which span multiple domains, including healthcare, finance, human activity recognition, and industrial monitoring. By outperforming 14 state-of-the-art methods, FIC-TSC demonstrates superior robustness to domain shifts—a crucial capability for real-world deployment.

In summary, this work sets a foundation for integrating Fisher information constraints into time series learning, with broad implications for both theory and practical applications.

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

## A. Proofs

**Lemma 1.** *At a local minimum, the expected Hessian matrix of the negative log-likelihood is asymptotically equivalent to the Fisher Information Matrix, w.r.t $\Theta$, which is presented as,*

$$\mathbb{E}_{p(\mathcal{D}|\Theta)}\left[\nabla^2(-\log p(\mathcal{D}\mid\Theta))\right]=\mathcal{F}. \tag{10}$$

*Proof.* We first obtain,

$$\mathbb{E}_{p(\mathcal{D}|\Theta)}\left[\nabla^2(-\log p(\mathcal{D}\mid\Theta))\right] \tag{11}$$
$$=-\mathbb{E}_{p(\mathcal{D}|\Theta)}\left[\nabla^2(\log p(\mathcal{D}\mid\Theta))\right].$$

Now, we focus on the term $\nabla^2\log p(\mathcal{D}\mid\Theta)$,

$$\nabla^2\log p(\mathcal{D}\mid\Theta) \tag{12}$$
$$=\nabla\left(\nabla\log p(\mathcal{D}\mid\Theta)\right)=\nabla\left(\frac{\nabla p(\mathcal{D}\mid\Theta)}{p(\mathcal{D}\mid\Theta)}\right)$$
$$=\frac{\nabla^2 p(\mathcal{D}\mid\Theta)p(\mathcal{D}\mid\Theta)-\nabla p(\mathcal{D}\mid\Theta)\nabla p(\mathcal{D}\mid\Theta)^{\mathrm{T}}}{p(\mathcal{D}\mid\Theta)p(\mathcal{D}\mid\Theta)}$$
$$=\frac{\nabla^2 p(\mathcal{D}\mid\Theta)p(\mathcal{D}\mid\Theta)}{p(\mathcal{D}\mid\Theta)p(\mathcal{D}\mid\Theta)}-\frac{\nabla p(\mathcal{D}\mid\Theta)\nabla p(\mathcal{D}\mid\Theta)^{\mathrm{T}}}{p(\mathcal{D}\mid\Theta)p(\mathcal{D}\mid\Theta)}$$
$$=\frac{\nabla^2 p(\mathcal{D}\mid\Theta)}{p(\mathcal{D}\mid\Theta)}-\left(\frac{\nabla p(\mathcal{D}\mid\Theta)}{p(\mathcal{D}\mid\Theta)}\right)\left(\frac{\nabla p(\mathcal{D}\mid\Theta)}{p(\mathcal{D}\mid\Theta)}\right)^{\mathrm{T}}.$$

At a local minimum, $\nabla p(\mathcal{D}\mid\Theta)=0$. Therefore,

$$\mathbb{E}_{p(\mathcal{D}|\Theta)}\left[\nabla^2\log p(\mathcal{D}\mid\Theta)\right] \tag{13}$$
$$=\int\frac{\nabla^2 p(\mathcal{D}\mid\Theta)}{p(\mathcal{D}\mid\Theta)}p(\mathcal{D}\mid\Theta)\mathrm{d}x$$
$$-\mathbb{E}_{p(\mathcal{D}|\Theta)}\left[\nabla\log p(\mathcal{D}\mid\Theta)\nabla\log p(\mathcal{D}\mid\Theta)^{\mathrm{T}}\right]$$
$$=\nabla^2\int p(\mathcal{D}\mid\Theta)\mathrm{d}x-\mathcal{F}$$
$$=-\mathcal{F}.$$

Substitute Eq. 13 to Eq. 11, the proof is completed. $\square$

**Corollary 1.** *At a local minimum, the upper bound of $\alpha$-sharpness is able to be approximated via Taylor expansion as*

$$\alpha\text{-sharpness}\propto\frac{\alpha^2\|\mathcal{F}\|_1}{2(1+\mathcal{L}(\Theta))}. \tag{14}$$

*Proof.* We first applying the Taylor expansion for $\mathcal{L}(\Theta')$,

$$\mathcal{L}(\Theta')\approx\mathcal{L}(\Theta)+\nabla\mathcal{L}(\Theta)^{\top}(\Theta'-\Theta)+ \tag{15}$$
$$+\frac{1}{2}(\Theta'-\Theta)^{\top}\nabla^2\mathcal{L}(\Theta)(\Theta'-\Theta)+\mathcal{O}(\Theta'-\Theta).$$

When $\Theta$ is a local minimum, $\nabla\mathcal{L}(\Theta)=0$, accordingly,

$$\mathcal{L}(\Theta')-\mathcal{L}(\Theta)=\frac{1}{2}(\Theta'-\Theta)^{\top}\nabla^2\mathcal{L}(\Theta)(\Theta'-\Theta). \tag{16}$$

In this quadratic form, we note that:

$$\frac{\max_{\Theta' \in B_2(\alpha,\Theta)} \left(L\left(\Theta'\right) - \mathcal{L}(\Theta)\right)}{1 + \mathcal{L}(\Theta)} = \frac{\alpha^2 \|\nabla^2 \mathcal{L}(\Theta)\|_2}{2(1 + \mathcal{L}(\Theta))}, \tag{17}$$

where $\|\nabla^2 \mathcal{L}(\Theta)\|_2$ denotes the spectral norm of the matrix, which is equal to its largest eigenvalue $\lambda_{max}$.

We also note, at a local minimum, $\nabla^2 \mathcal{L}(\Theta)$ is positive semidefinite, hence,

$$\|\texttt{diag}(\nabla^2 \mathcal{L}(\Theta))\|_1 = \sum_i \lambda_i \geq \lambda_{max}, \tag{18}$$

therefore,

$$\frac{\max_{\Theta' \in B_2(\alpha,\Theta)} \left(L\left(\Theta'\right) - \mathcal{L}(\Theta)\right)}{1 + \mathcal{L}(\Theta)} = \frac{\alpha^2 \|\nabla^2 \mathcal{L}(\Theta)\|_2}{2(1 + \mathcal{L}(\Theta))} \leq \frac{\alpha^2 \|\nabla^2 \mathcal{L}(\Theta)\|_1}{2(1 + \mathcal{L}(\Theta))}. \tag{19}$$

Under the diagonal approximation of FIM, after applying Lemma 1 above, the proof is completed. □

**Theorem 1.** *Consider a L-Lipschitz objective function $\mathcal{L}(\Theta)$, defined such that for any two points $\Theta_1$ and $\Theta_2$, the inequality $\|\nabla \mathcal{L}(\Theta_1) - \nabla \mathcal{L}(\Theta_2)\| \leq L\|\Theta_1 - \Theta_2\|$ holds. By appropriately choosing the learning rate $\eta$ and the FIC constraint $\epsilon$, the convergence rate of gradient descent can be expressed as $\mathcal{O}(\frac{1}{T})$, where $T$ represents the number of iterations.*

*Proof.* We first reiterate the update rule at iteration $t$ via gradient descent,

$$\Theta_{t+1} = \Theta_t - \eta \nabla \mathcal{L}(\Theta_t)', \tag{20}$$

where $\eta$ denotes the learning rate, and $\nabla \mathcal{L}(\Theta_t)'$ denotes the gradient constrained by FIC. The Lipschitz continuity assumption can bound the change in the loss function:

$$\mathcal{L}(\Theta_{t+1}) \leq \mathcal{L}(\Theta_t) + \nabla \mathcal{L}(\Theta_t)^\top (\Theta_{t+1} - \Theta_t) + \frac{L}{2}\|\Theta_{t+1} - \Theta_t\|^2. \tag{21}$$

Substitute Eq. 20 to above equation, and after simplifying and rearranging,

$$\mathcal{L}(\Theta_{t+1}) - \mathcal{L}(\Theta_t) \leq -\eta \nabla \mathcal{L}(\Theta_t)^\top \nabla \mathcal{L}(\Theta_t)' + \frac{L\epsilon\eta^2}{2}. \tag{22}$$

Since $\nabla \mathcal{L}(\Theta_t)^\top \nabla \mathcal{L}(\Theta_t)' \geq 0$ always holds, when selecting a suitable $\epsilon$ and $\eta$, we can ensure the loss is **sufficiently decreasing**. Subsequently, when the model at $T$th iteration from a initial point $\Theta_1$,

$$\mathcal{L}(\Theta_T) - \mathcal{L}(\Theta_1) \leq \sum_{t=1}^{T} \left( -\eta \nabla \mathcal{L}(\Theta_t)^\top \nabla \mathcal{L}(\Theta_t)' + \frac{L\epsilon\eta^2}{2} \right). \tag{23}$$

Suppose a stationary point[3] exists during these $T$ iterations, which is denoted as $\Theta^*$. The following inequalities hold,

$$\mathcal{L}(\Theta^*) - \mathcal{L}(\Theta_1) \leq \mathcal{L}(\Theta_T) - \mathcal{L}(\Theta_1), \tag{24}$$

and

$$\|\nabla \mathcal{L}(\Theta^*)\|^2 = \min_{t \in [T]} \|\nabla \mathcal{L}(\Theta_t)\|^2, \tag{25}$$

---

[3]At this stage, for simplicity, we assume it is a local minimum.

where $[T] = \{1, ..., T\}$. Substitute Eq. 24 to Eq. 23 and rearrange,

$$\frac{1}{T}\sum_{t=1}^{T} \nabla\mathcal{L}(\Theta_t)^\top \nabla\mathcal{L}(\Theta_t)' \leq \frac{\mathcal{L}(\Theta_1) - \mathcal{L}(\Theta^*)}{\eta T} + \frac{L\epsilon\eta}{2}. \tag{26}$$

We also have the below inequality,

$$\min_{t \in [T]} \nabla\mathcal{L}(\Theta_t)^\top \nabla\mathcal{L}(\Theta_t)' \leq \frac{1}{T}\sum_{t=1}^{T} \nabla\mathcal{L}(\Theta_t)^\top \nabla\mathcal{L}(\Theta_t)'. \tag{27}$$

We can select a suitable $\epsilon$ (i.e. not too small) such that

$$\min_{t \in [T]} \|\nabla\mathcal{L}(\Theta_t)\|^2 = \min_{t \in [T]} \nabla\mathcal{L}(\Theta_t)^\top \nabla\mathcal{L}(\Theta_t)'. \tag{28}$$

Consequently, according to Eqs. 25, 26, 27, and 28, we can immediately obtain a convergence rate of $O(1/T)$, which demonstrates a sublinear rate of convergence. □

## B. Algorithm Detail

Please refer to Algorithm 1.

---
**Algorithm 1** FIC-TS (Training Phase)
---
**Input:** Initial Neural network $\Theta$, Optimizer, number of iterations $T$, loss function $\mathcal{L}$, dataset $\mathcal{S}$
**Output:** Trained network $\Theta$
1: **for** $t = 1$ to $T$ **do**
2:     Sample a mini-batch of data $(\mathcal{B}_X, \mathcal{B}_Y) \in \mathcal{S}$
3:     Forward pass $\hat{\mathcal{B}_Y}_i = f(\mathcal{B}_X; \theta)$
4:     Compute loss $\mathcal{L}(\hat{\mathcal{B}_Y}, \mathcal{B}_Y)$
5:     Backward pass $\nabla_\theta\mathcal{L}(\hat{\mathcal{B}_Y}, \mathcal{B}_Y)$
6:     Compute $\|\mathcal{F}\|$
7:     **if** $\|\mathcal{F}\| \geq \epsilon$ **then**
8:         $\nabla\mathcal{L}(\Theta) \leftarrow \sqrt{\frac{\epsilon}{\|\mathcal{F}\|}}\nabla\mathcal{L}(\Theta)$
9:     **end if**
10:    Update parameters `Optimizer.step()`
11: **end for**

---

## C. Dataset Description

**UEA 30 Datasets.** The detail of all 30 datasets is provided in Table S7. It should be noteworthy that the datasets `JapaneseVowels` and `SpokenArabicDigits` used in the Group 2 Experiment originally have varied lengths of sequences. We pre-process data following (Wu et al., 2023a), where we pad them to 29 and 93, respectively.

**UCR dataset.** We present the full list of UCR datasets in Fig. S10.

## D. Implementation Detail

We use AdamW optimizer (Loshchilov & Hutter, 2017) with a learning rate of 5e-3 and a weight decay of 1e-4. We implemented all experiments on a cluster node with NVIDIA A100 (40 GB). We use Pytorch Library (Paszke et al., 2019) with version of 1.13. we implement our algorithm on a network based on (Ismail Fawaz et al., 2020). The architecture of the network can be simply presented as:

$$\boldsymbol{X} \in \mathbb{R}^{d \times T} \xrightarrow{f_{\text{feat}}} \boldsymbol{X} \in \mathbb{R}^{128 \times T} \xrightarrow{mean\ pool.} \boldsymbol{X} \in \mathbb{R}^{128} \xrightarrow{f_{\text{MLP}}} \hat{\boldsymbol{Y}} \in \mathbb{R}^C, \tag{29}$$

*Table S7.* Dataset Summary

| Dataset | Training Size | Test Size | Dimensions | Length | Classes |
|---|---|---|---|---|---|
| ArticularyWordRecognition | 275 | 300 | 9 | 144 | 25 |
| AtrialFibrillation | 15 | 15 | 2 | 640 | 3 |
| BasicMotions | 40 | 40 | 6 | 100 | 4 |
| CharacterTrajectories | 1422 | 1436 | 3 | 182 | 20 |
| Cricket | 108 | 72 | 6 | 1197 | 12 |
| DuckDuckGeese | 60 | 40 | 1345 | 270 | 5 |
| EigenWorms | 128 | 131 | 6 | 17984 | 5 |
| Epilepsy | 137 | 138 | 3 | 206 | 4 |
| EthanolConcentration | 261 | 263 | 3 | 1751 | 4 |
| ERing | 30 | 30 | 4 | 65 | 6 |
| FaceDetection | 5890 | 3524 | 144 | 62 | 2 |
| FingerMovements | 316 | 100 | 28 | 50 | 2 |
| HandMovementDirection | 320 | 147 | 10 | 400 | 4 |
| Handwriting | 150 | 850 | 3 | 152 | 26 |
| Heartbeat | 204 | 205 | 61 | 405 | 2 |
| JapaneseVowels | 270 | 370 | 12 | 29 (max) | 9 |
| Libras | 180 | 180 | 2 | 45 | 15 |
| LSST | 2459 | 2466 | 6 | 36 | 14 |
| InsectWingbeat | 30000 | 20000 | 200 | 78 | 10 |
| MotorImagery | 278 | 100 | 64 | 3000 | 2 |
| NATOPS | 180 | 180 | 24 | 51 | 6 |
| PenDigits | 7494 | 3498 | 2 | 8 | 10 |
| PEMS-SF | 267 | 173 | 963 | 144 | 7 |
| Phoneme | 3315 | 3353 | 11 | 217 | 39 |
| RacketSports | 151 | 152 | 6 | 30 | 4 |
| SelfRegulationSCP1 | 268 | 293 | 6 | 896 | 2 |
| SelfRegulationSCP2 | 200 | 180 | 7 | 1152 | 2 |
| SpokenArabicDigits | 6599 | 2199 | 13 | 93 (max) | 10 |
| StandWalkJump | 12 | 15 | 4 | 2500 | 3 |
| UWaveGestureLibrary | 120 | 320 | 3 | 315 | 8 |

```
Adiac, ArrowHead, Beef, BeetleFly, BirdChicken, Car, CBF, ChlorineConcentration,
CinCECGTorso, Coffee, Computers, CricketX, CricketY, CricketZ, DiatomSizeReduction,
DistalPhalanxOutlineAgeGroup, DistalPhalanxOutlineCorrect, DistalPhalanxTW,
Earthquakes, ECG200, ECG5000, ECGFiveDays, ElectricDevices, FaceAll, FaceFour,
FacesUCR, FiftyWords, Fish, FordA, FordB, GunPoint, Ham, HandOutlines,
Haptics, Herring, InlineSkate, InsectWingbeatSound, ItalyPowerDemand,
LargeKitchenAppliances, Lightning2, Lightning7, Mallat, Meat, MedicalImages,
MiddlePhalanxOutlineAgeGroup, MiddlePhalanxOutlineCorrect, MiddlePhalanxTW,
MoteStrain, NonInvasiveFetalECGThorax1, NonInvasiveFetalECGThorax2, OliveOil,
OSULeaf, PhalangesOutlinesCorrect, Phoneme, Plane, ProximalPhalanxOutlineAgeGroup,
ProximalPhalanxOutlineCorrect, ProximalPhalanxTW, RefrigerationDevices,
ScreenType, ShapeletSim, ShapesAll, SmallKitchenAppliances, SonyAIBORobotSurface1,
SonyAIBORobotSurface2, StarLightCurves, Strawberry, SwedishLeaf, Symbols,
SyntheticControl, ToeSegmentation1, ToeSegmentation2, Trace, TwoLeadECG,
TwoPatterns, UWaveGestureLibraryAll, UWaveGestureLibraryX, UWaveGestureLibraryY,
UWaveGestureLibraryZ, Wafer, Wine, WordSynonyms, Worms, WormsTwoClass, Yoga.
```

*Figure S10.* Full list of UCR 85 datasets.

where $f_{\texttt{feat}}$ denotes the backbone feature extractor, following the specifications detailed by (Ismail Fawaz et al., 2020). The MLP-based classifier, denoted as $f_{\texttt{MLP}}$, comprises two sequential layers: the first layer features $128 \times 128$ neurons with ReLu activation function, and the second layer, designed to output class probabilities, includes $128 \times C$ neurons, where $C$ represents the number of classes.

*Table S8.* A summary of the Wilcoxon signed-rank test on Effectiveness of RevIN.

| Comparison | p-Value |
|---|---|
| **w/** RevIN vs **w/o** RevIN | 0.889 |

*Table S9.* A summary of the Wilcoxon signed-rank test on Ablation Analysis.

| Comparison | p-Value |
|---|---|
| Itime vs Itime+FIC | 0.001 |

## E. Additional Statistical Test

As suggested by (Demšar, 2006), we can conduct the Wilcoxon signed-rank test to compare the performance of two classifiers across different datasets.

### E.1. Statistical Test on Effectiveness of RevIN

Please refer to Table S8, which indicates that there is no significant difference in classification performance between using `RevIN` and not using `RevIN`. This suggests that `RevIN` is not helpful for classification.

### E.2. Statistical Test on Ablation Analysis

Please refer to Table S9. Due to the p-value being tiny and much smaller than 0.05, we have the confidence to conclude that our method is statistically superior to the baseline.

### E.3. Statistical Test on SAM and FIC

Please refer to Table S10.

## F. Full Results

### F.1. Multivariate Time Series Classification

Please refer to Table S11 below.

### F.2. Univariate Time Series Classification

Please refer to Table S12 below.

*Table S10.* A summary of the Wilcoxon signed-rank test on the comparison between SAM and FIC.

| Comparison | p-Value |
|---|---|
| Accuracy | 0.006 |
| Runtime | 0.001 |

*Table S11.* The full results on 30 UEA datasets. We reported multiple metrics, including Accuracy, balanced Accuracy, F1, Precision (P), and Recall (R).

| Dataset | Uni. | | | | | Full | | | | |
|---|---|---|---|---|---|---|---|---|---|---|
| | Accuracy | Bal. Accuracy | F1 marco | P marco | R marco | Accuracy | Bal. Accuracy | F1 marco | P marco | R marco |
| ArticularyWordRecognition | 0.993 | 0.993 | 0.993 | 0.994 | 0.993 | 0.998 | 0.998 | 0.998 | 0.998 | 0.998 |
| AtrialFibrillation | 0.567 | 0.567 | 0.505 | 0.486 | 0.567 | 0.600 | 0.600 | 0.522 | 0.477 | 0.600 |
| BasicMotions | 1.000 | 1.000 | 1.000 | 1.000 | 1.000 | 1.000 | 1.000 | 1.000 | 1.000 | 1.000 |
| CharacterTrajectories | 0.997 | 0.997 | 0.997 | 0.997 | 0.997 | 0.999 | 0.998 | 0.998 | 0.999 | 0.998 |
| Cricket | 1.000 | 1.000 | 1.000 | 1.000 | 1.000 | 1.000 | 1.000 | 1.000 | 1.000 | 1.000 |
| DuckDuckGeese | 0.650 | 0.650 | 0.637 | 0.694 | 0.650 | 0.720 | 0.720 | 0.718 | 0.769 | 0.720 |
| EigenWorms | 0.855 | 0.797 | 0.804 | 0.850 | 0.797 | 0.924 | 0.896 | 0.893 | 0.892 | 0.896 |
| ERing | 0.919 | 0.919 | 0.919 | 0.924 | 0.919 | 0.954 | 0.954 | 0.954 | 0.955 | 0.954 |
| Epilepsy | 0.989 | 0.990 | 0.989 | 0.989 | 0.990 | 1.000 | 1.000 | 1.000 | 1.000 | 1.000 |
| EthanolConcentration | 0.392 | 0.392 | 0.386 | 0.402 | 0.392 | 0.401 | 0.401 | 0.395 | 0.418 | 0.401 |
| FaceDetection | 0.684 | 0.684 | 0.684 | 0.685 | 0.684 | 0.692 | 0.692 | 0.691 | 0.692 | 0.692 |
| FingerMovements | 0.650 | 0.647 | 0.639 | 0.666 | 0.647 | 0.715 | 0.716 | 0.714 | 0.718 | 0.716 |
| HandMovementDirection | 0.493 | 0.437 | 0.443 | 0.506 | 0.437 | 0.547 | 0.514 | 0.517 | 0.584 | 0.514 |
| Handwriting | 0.616 | 0.613 | 0.591 | 0.647 | 0.613 | 0.730 | 0.725 | 0.714 | 0.740 | 0.725 |
| Heartbeat | 0.810 | 0.715 | 0.735 | 0.780 | 0.715 | 0.820 | 0.746 | 0.760 | 0.784 | 0.746 |
| InsectWingbeat | 0.711 | 0.711 | 0.711 | 0.713 | 0.711 | 0.723 | 0.723 | 0.721 | 0.722 | 0.723 |
| JapaneseVowels | 0.991 | 0.989 | 0.990 | 0.990 | 0.989 | 0.996 | 0.996 | 0.996 | 0.995 | 0.996 |
| Libras | 0.794 | 0.794 | 0.790 | 0.811 | 0.794 | 0.917 | 0.917 | 0.916 | 0.923 | 0.917 |
| LSST | 0.653 | 0.449 | 0.461 | 0.597 | 0.449 | 0.668 | 0.443 | 0.461 | 0.581 | 0.443 |
| MotorImagery | 0.650 | 0.650 | 0.649 | 0.651 | 0.650 | 0.685 | 0.685 | 0.683 | 0.690 | 0.685 |
| NATOPS | 0.989 | 0.989 | 0.989 | 0.989 | 0.989 | 0.992 | 0.992 | 0.992 | 0.992 | 0.992 |
| PEMS-SF | 0.792 | 0.797 | 0.788 | 0.796 | 0.797 | 0.879 | 0.878 | 0.875 | 0.883 | 0.878 |
| PenDigits | 0.976 | 0.976 | 0.976 | 0.978 | 0.976 | 0.985 | 0.985 | 0.985 | 0.986 | 0.985 |
| PhonemeSpectra | 0.313 | 0.313 | 0.300 | 0.318 | 0.313 | 0.323 | 0.323 | 0.314 | 0.330 | 0.323 |
| RacketSports | 0.898 | 0.906 | 0.904 | 0.905 | 0.906 | 0.908 | 0.915 | 0.914 | 0.915 | 0.915 |
| SelfRegulationSCP1 | 0.901 | 0.901 | 0.901 | 0.901 | 0.901 | 0.903 | 0.903 | 0.903 | 0.904 | 0.903 |
| SelfRegulationSCP2 | 0.594 | 0.594 | 0.589 | 0.599 | 0.594 | 0.608 | 0.608 | 0.608 | 0.609 | 0.608 |
| StandWalkJump | 0.633 | 0.633 | 0.620 | 0.640 | 0.633 | 0.667 | 0.667 | 0.631 | 0.715 | 0.667 |
| SpokenArabicDigits | 1.000 | 1.000 | 1.000 | 1.000 | 1.000 | 1.000 | 1.000 | 1.000 | 1.000 | 1.000 |
| UWaveGestureLibrary | 0.902 | 0.902 | 0.900 | 0.906 | 0.902 | 0.930 | 0.930 | 0.929 | 0.933 | 0.930 |
| Avg. | 0.780 | 0.767 | 0.763 | 0.780 | 0.767 | 0.809 | 0.797 | 0.793 | 0.807 | 0.797 |

*Table S12.* The full results on 85 UCR datasets. We reported multiple metrics, including Accuracy, balanced Accuracy, F1, Precision (P), and Recall (R).

| Dataset | Uni. | | | | | Full | | | | |
|---|---|---|---|---|---|---|---|---|---|---|
| | Accuracy | Bal. Accuracy | F1 marco | P marco | R marco | Accuracy | Bal. Accuracy | F1 marco | P marco | R marco |
| Adiac | 0.778 | 0.780 | 0.760 | 0.792 | 0.780 | 0.786 | 0.791 | 0.774 | 0.812 | 0.791 |
| ArrowHead | 0.909 | 0.908 | 0.906 | 0.908 | 0.908 | 0.911 | 0.908 | 0.909 | 0.913 | 0.908 |
| Beef | 0.800 | 0.800 | 0.798 | 0.825 | 0.800 | 0.833 | 0.833 | 0.828 | 0.866 | 0.833 |
| BeetleFly | 1.000 | 1.000 | 1.000 | 1.000 | 1.000 | 1.000 | 1.000 | 1.000 | 1.000 | 1.000 |
| BirdChicken | 1.000 | 1.000 | 1.000 | 1.000 | 1.000 | 1.000 | 1.000 | 1.000 | 1.000 | 1.000 |
| Car | 0.925 | 0.914 | 0.920 | 0.942 | 0.914 | 0.933 | 0.924 | 0.931 | 0.952 | 0.924 |
| CBF | 1.000 | 1.000 | 1.000 | 1.000 | 1.000 | 1.000 | 1.000 | 1.000 | 1.000 | 1.000 |
| ChlorineConcentration | 0.835 | 0.806 | 0.814 | 0.825 | 0.806 | 0.847 | 0.804 | 0.823 | 0.859 | 0.804 |
| CinCECGTorso | 0.785 | 0.785 | 0.782 | 0.796 | 0.785 | 0.811 | 0.811 | 0.808 | 0.819 | 0.811 |
| Coffee | 1.000 | 1.000 | 1.000 | 1.000 | 1.000 | 1.000 | 1.000 | 1.000 | 1.000 | 1.000 |
| Computers | 0.834 | 0.834 | 0.834 | 0.834 | 0.834 | 0.880 | 0.880 | 0.880 | 0.880 | 0.880 |
| CricketX | 0.841 | 0.843 | 0.840 | 0.850 | 0.843 | 0.854 | 0.857 | 0.853 | 0.862 | 0.857 |
| CricketY | 0.826 | 0.827 | 0.826 | 0.834 | 0.827 | 0.844 | 0.845 | 0.843 | 0.848 | 0.845 |
| CricketZ | 0.850 | 0.842 | 0.841 | 0.849 | 0.842 | 0.862 | 0.854 | 0.855 | 0.863 | 0.854 |
| DiatomSizeReduction | 0.979 | 0.961 | 0.968 | 0.978 | 0.961 | 0.990 | 0.982 | 0.987 | 0.992 | 0.982 |
| DistalPhalanxOutlineAgeGroup | 0.788 | 0.789 | 0.796 | 0.805 | 0.789 | 0.806 | 0.779 | 0.806 | 0.850 | 0.779 |
| DistalPhalanxOutlineCorrect | 0.812 | 0.790 | 0.798 | 0.823 | 0.790 | 0.819 | 0.809 | 0.812 | 0.818 | 0.809 |
| DistalPhalanxTW | 0.745 | 0.577 | 0.570 | 0.604 | 0.577 | 0.748 | 0.610 | 0.586 | 0.588 | 0.610 |
| Earthquakes | 0.791 | 0.633 | 0.640 | 0.807 | 0.633 | 0.802 | 0.664 | 0.684 | 0.767 | 0.664 |
| ECG200 | 0.930 | 0.915 | 0.923 | 0.932 | 0.915 | 0.935 | 0.925 | 0.929 | 0.933 | 0.925 |
| ECG5000 | 0.945 | 0.554 | 0.591 | 0.675 | 0.554 | 0.946 | 0.569 | 0.616 | 0.722 | 0.569 |
| ECGFiveDays | 1.000 | 1.000 | 1.000 | 1.000 | 1.000 | 1.000 | 1.000 | 1.000 | 1.000 | 1.000 |
| ElectricDevices | 0.740 | 0.630 | 0.629 | 0.671 | 0.630 | 0.788 | 0.722 | 0.721 | 0.753 | 0.722 |
| FaceAll | 0.934 | 0.949 | 0.925 | 0.913 | 0.949 | 0.959 | 0.953 | 0.946 | 0.943 | 0.953 |
| FaceFour | 0.955 | 0.958 | 0.959 | 0.962 | 0.958 | 0.966 | 0.968 | 0.970 | 0.974 | 0.968 |
| FacesUCR | 0.957 | 0.940 | 0.943 | 0.948 | 0.940 | 0.960 | 0.944 | 0.946 | 0.949 | 0.944 |
| FiftyWords | 0.803 | 0.678 | 0.673 | 0.707 | 0.678 | 0.813 | 0.685 | 0.676 | 0.703 | 0.685 |
| Fish | 0.991 | 0.993 | 0.992 | 0.991 | 0.993 | 0.997 | 0.998 | 0.997 | 0.997 | 0.998 |
| FordA | 0.966 | 0.966 | 0.966 | 0.966 | 0.966 | 0.967 | 0.967 | 0.967 | 0.967 | 0.967 |
| FordB | 0.862 | 0.862 | 0.862 | 0.862 | 0.862 | 0.869 | 0.869 | 0.869 | 0.869 | 0.869 |
| GunPoint | 1.000 | 1.000 | 1.000 | 1.000 | 1.000 | 1.000 | 1.000 | 1.000 | 1.000 | 1.000 |
| Ham | 0.824 | 0.826 | 0.823 | 0.834 | 0.826 | 0.843 | 0.844 | 0.843 | 0.845 | 0.844 |
| HandOutlines | 0.958 | 0.947 | 0.954 | 0.963 | 0.947 | 0.970 | 0.964 | 0.968 | 0.971 | 0.964 |
| Haptics | 0.528 | 0.528 | 0.518 | 0.542 | 0.528 | 0.534 | 0.535 | 0.523 | 0.553 | 0.535 |
| Herring | 0.742 | 0.698 | 0.697 | 0.798 | 0.698 | 0.750 | 0.723 | 0.722 | 0.778 | 0.723 |
| InlineSkate | 0.356 | 0.367 | 0.361 | 0.381 | 0.367 | 0.384 | 0.395 | 0.386 | 0.395 | 0.395 |
| InsectWingbeatSound | 0.627 | 0.627 | 0.618 | 0.637 | 0.627 | 0.641 | 0.641 | 0.630 | 0.641 | 0.641 |
| ItalyPowerDemand | 0.974 | 0.974 | 0.974 | 0.974 | 0.974 | 0.976 | 0.976 | 0.976 | 0.976 | 0.976 |
| LargeKitchenAppliances | 0.917 | 0.917 | 0.917 | 0.919 | 0.917 | 0.932 | 0.932 | 0.932 | 0.932 | 0.932 |
| Lightning2 | 0.918 | 0.919 | 0.918 | 0.918 | 0.919 | 0.926 | 0.926 | 0.926 | 0.926 | 0.926 |
| Lightning7 | 0.897 | 0.910 | 0.896 | 0.896 | 0.910 | 0.918 | 0.923 | 0.915 | 0.924 | 0.923 |
| Mallat | 0.970 | 0.970 | 0.970 | 0.970 | 0.970 | 0.980 | 0.980 | 0.980 | 0.980 | 0.980 |
| Meat | 0.942 | 0.942 | 0.942 | 0.950 | 0.942 | 0.958 | 0.958 | 0.958 | 0.959 | 0.958 |
| MedicalImages | 0.808 | 0.751 | 0.761 | 0.789 | 0.751 | 0.818 | 0.791 | 0.792 | 0.808 | 0.791 |
| MiddlePhalanxOutlineAgeGroup | 0.666 | 0.481 | 0.497 | 0.810 | 0.481 | 0.679 | 0.511 | 0.538 | 0.789 | 0.511 |
| MiddlePhalanxOutlineCorrect | 0.869 | 0.858 | 0.864 | 0.876 | 0.858 | 0.878 | 0.872 | 0.875 | 0.879 | 0.872 |
| MiddlePhalanxTW | 0.627 | 0.435 | 0.414 | 0.429 | 0.435 | 0.633 | 0.468 | 0.450 | 0.470 | 0.468 |
| MoteStrain | 0.925 | 0.925 | 0.924 | 0.924 | 0.925 | 0.928 | 0.928 | 0.927 | 0.927 | 0.928 |
| NonInvasiveFetalECGThorax1 | 0.927 | 0.926 | 0.923 | 0.933 | 0.926 | 0.936 | 0.935 | 0.934 | 0.937 | 0.935 |
| NonInvasiveFetalECGThorax2 | 0.936 | 0.934 | 0.932 | 0.937 | 0.934 | 0.944 | 0.940 | 0.938 | 0.942 | 0.940 |
| OliveOil | 0.800 | 0.719 | 0.672 | 0.714 | 0.719 | 0.833 | 0.764 | 0.744 | 0.858 | 0.764 |
| OSULeaf | 0.936 | 0.922 | 0.927 | 0.938 | 0.922 | 0.948 | 0.930 | 0.939 | 0.955 | 0.930 |
| PhalangesOutlinesCorrect | 0.857 | 0.839 | 0.846 | 0.857 | 0.839 | 0.860 | 0.840 | 0.848 | 0.862 | 0.840 |
| Phoneme | 0.332 | 0.186 | 0.188 | 0.228 | 0.186 | 0.350 | 0.237 | 0.235 | 0.278 | 0.237 |
| Plane | 1.000 | 1.000 | 1.000 | 1.000 | 1.000 | 1.000 | 1.000 | 1.000 | 1.000 | 1.000 |
| ProximalPhalanxOutlineAgeGroup | 0.885 | 0.787 | 0.806 | 0.836 | 0.787 | 0.893 | 0.825 | 0.841 | 0.861 | 0.825 |
| ProximalPhalanxOutlineCorrect | 0.931 | 0.915 | 0.920 | 0.925 | 0.915 | 0.940 | 0.920 | 0.929 | 0.941 | 0.920 |
| ProximalPhalanxTW | 0.834 | 0.575 | 0.585 | 0.671 | 0.575 | 0.834 | 0.550 | 0.539 | 0.547 | 0.550 |
| RefrigerationDevices | 0.603 | 0.603 | 0.597 | 0.608 | 0.603 | 0.611 | 0.611 | 0.602 | 0.612 | 0.611 |
| ScreenType | 0.628 | 0.628 | 0.626 | 0.637 | 0.628 | 0.637 | 0.637 | 0.635 | 0.648 | 0.637 |
| ShapeletSim | 1.000 | 1.000 | 1.000 | 1.000 | 1.000 | 1.000 | 1.000 | 1.000 | 1.000 | 1.000 |
| ShapesAll | 0.880 | 0.880 | 0.878 | 0.895 | 0.880 | 0.883 | 0.883 | 0.881 | 0.902 | 0.883 |
| SmallKitchenAppliances | 0.825 | 0.825 | 0.826 | 0.830 | 0.825 | 0.845 | 0.845 | 0.847 | 0.855 | 0.845 |
| SonyAIBORobotSurface1 | 0.958 | 0.957 | 0.957 | 0.956 | 0.957 | 0.973 | 0.974 | 0.973 | 0.972 | 0.974 |
| SonyAIBORobotSurface2 | 0.955 | 0.957 | 0.953 | 0.949 | 0.957 | 0.962 | 0.963 | 0.960 | 0.957 | 0.963 |
| StarLightCurves | 0.980 | 0.961 | 0.970 | 0.979 | 0.961 | 0.981 | 0.962 | 0.972 | 0.983 | 0.962 |
| Strawberry | 0.976 | 0.976 | 0.974 | 0.971 | 0.976 | 0.978 | 0.977 | 0.976 | 0.976 | 0.977 |
| SwedishLeaf | 0.964 | 0.965 | 0.964 | 0.965 | 0.965 | 0.965 | 0.965 | 0.965 | 0.966 | 0.965 |
| Symbols | 0.952 | 0.953 | 0.952 | 0.953 | 0.953 | 0.974 | 0.975 | 0.974 | 0.975 | 0.975 |
| SyntheticControl | 1.000 | 1.000 | 1.000 | 1.000 | 1.000 | 1.000 | 1.000 | 1.000 | 1.000 | 1.000 |
| ToeSegmentation1 | 0.974 | 0.974 | 0.974 | 0.974 | 0.974 | 0.976 | 0.977 | 0.976 | 0.976 | 0.977 |
| ToeSegmentation2 | 0.985 | 0.983 | 0.975 | 0.968 | 0.983 | 0.985 | 0.974 | 0.974 | 0.974 | 0.974 |
| Trace | 1.000 | 1.000 | 1.000 | 1.000 | 1.000 | 1.000 | 1.000 | 1.000 | 1.000 | 1.000 |
| TwoLeadECG | 1.000 | 1.000 | 1.000 | 1.000 | 1.000 | 1.000 | 1.000 | 1.000 | 1.000 | 1.000 |
| TwoPatterns | 1.000 | 1.000 | 1.000 | 1.000 | 1.000 | 1.000 | 1.000 | 1.000 | 1.000 | 1.000 |
| UWaveGestureLibraryAll | 0.845 | 0.846 | 0.841 | 0.854 | 0.846 | 0.846 | 0.847 | 0.838 | 0.854 | 0.847 |
| UWaveGestureLibraryX | 0.798 | 0.796 | 0.789 | 0.787 | 0.796 | 0.800 | 0.797 | 0.792 | 0.789 | 0.797 |
| UWaveGestureLibraryY | 0.667 | 0.669 | 0.665 | 0.683 | 0.669 | 0.691 | 0.692 | 0.692 | 0.697 | 0.692 |
| UWaveGestureLibraryZ | 0.706 | 0.708 | 0.701 | 0.709 | 0.708 | 0.721 | 0.722 | 0.712 | 0.726 | 0.722 |
| Wafer | 0.999 | 0.998 | 0.997 | 0.995 | 0.998 | 0.999 | 0.999 | 0.998 | 0.998 | 0.999 |
| Wine | 0.759 | 0.759 | 0.752 | 0.799 | 0.759 | 0.843 | 0.843 | 0.841 | 0.851 | 0.843 |
| WordSynonyms | 0.683 | 0.523 | 0.537 | 0.600 | 0.523 | 0.705 | 0.566 | 0.574 | 0.629 | 0.566 |
| Worms | 0.844 | 0.819 | 0.819 | 0.831 | 0.819 | 0.877 | 0.851 | 0.861 | 0.884 | 0.851 |
| WormsTwoClass | 0.857 | 0.847 | 0.851 | 0.864 | 0.847 | 0.883 | 0.877 | 0.880 | 0.884 | 0.877 |
| Yoga | 0.848 | 0.847 | 0.847 | 0.847 | 0.847 | 0.893 | 0.892 | 0.892 | 0.892 | 0.892 |
| Avg. | 0.862 | 0.834 | 0.833 | 0.852 | 0.834 | 0.873 | 0.848 | 0.848 | 0.864 | 0.848 |

# G. Discussion

### G.1. Why Choose ITime, PatchTST, and Timesnet as baselines?

The selection is based on their impressive impact on the TS community.

### G.2. Why choose 10 datasets in the Model Analysis Section?

These are widely chosen in different works, such as TimesNet (Wu et al., 2023a).

### G.3. Why is Our Method Potentially Better than SAM in TSC?

As SAM is not central to our work, we defer its detailed theoretical analysis to future work while providing insights for this superiority. We conjecture the following two reasons as following:

**Training sample size.** Most SAM papers focus on image datasets (Foret et al., 2020), which typically have large training sizes (60k 14M). In contrast, TSC datasets often have very limited training sizes, e.g., the UW and SCP2 datasets have only 120 and 200 training samples, respectively. SAM's effectiveness in such cases remains unexplored.

**Sensitive to hyper-parameters.** As studied in (Andriushchenko & Flammarion, 2022) (e.g., Fig.16), SAM's performance is highly sensitive to its hyper-parameters (e.g., dataset-dependent batch size and perturbation radius). Poor choices can easily lead to worse performance than standard training. Given diverse TSC datasets, identifying universal hyper-parameters for SAM performing well on most datasets is challenging.

### G.4. Is Diagonal Approximation Necessary?

**Yes.** DL models typically have huge parameters (e.g., ITime has 600k parameters on SCP1, so Transformer-based models can have even more parameters). Therefore, computing and storing the full FIM on a typical GPU is extremely inefficient or even not feasible. We have also tested it on an A100 GPU, and the results support this claim. We also found related works (Kirkpatrick et al., 2017; Lee et al., 2017; Jhunjhunwala et al., 2024) that consistently apply diagonal approximation to tackle a similar computational issue, and they mention that the diagonal elements contain sufficiently important information.

It is worth mentioning that EWC (Lee et al., 2017) preserves prior knowledge by penalizing changes to important weights, using a Gaussian posterior centered at previous weights with precision from the observed Fisher information (Laplace approximation). Notably, EWC uses a diagonal approximation, aligning with and supporting the efficiency goals of our work.

Another relevant work, K-FAC (Martens & Grosse, 2015), addresses the high computational cost of the FIM by approximating large blocks of it, corresponding to entire layers, as the Kronecker product of two much smaller matrices. We consider this a promising direction for future work to achieve more accurate and efficient FIM approximations.

### G.5. Is Analysis of Non-Minimum Points Needed?

**No.** Our theoretical analysis aims to deliver the **achievability** of a better convergence. Since the optimizers [by simply adjusting hyper-parameters], in general, can easily reach local minima after convergence in the TSC task, it is sufficient to evaluate sharpness at local minima and their neighbors. In Proposition 1, we only claim that an appropriate FIC could potentially lead to a convergence to flatter minima. Hence, non-minimum points do not affect our conclusion regarding achievability.

This focus on achievability is analogous to the approach commonly used in *Coding Theory* (Cover, 1999), where initial results often emphasize achievable rates to demonstrate feasibility before refining practical implementation further. Similarly, our work lays the groundwork for future exploration of theoretical optimality.

Moreover, this claim is strongly supported by empirical evidence, where our method achieves $\sim 40\%$ reduction in sharpness and $\sim 4\%$ gain in accuracy as presented in Figs. 5, 8, and 9. These results validate the practical implications of our theoretical analysis and demonstrate the effectiveness of our proposed approach.

### G.6. Can FIC Compete Related Methods Related to Domain Shift Problem?

**Yes.** We compare two related methods that target to solve domain-shift problem:

**RevIN.** Our method outperforms RevIN, which is the most common approach for addressing domain shift in time series. The main motivation of our work is that RevIN's effectiveness in time series classification remains unexplored. Accordingly, we conducted an empirical investigation that demonstrates RevIN's ineffectiveness (see Sec. 3).

**SAM.** Our method outperforms SAM in both accuracy and efficiency in TSC, as presented in Table 5.

### G.7. Sharpness and Generalization

While the link between sharpness and generalization is out of our focus, here we want to include more discussion about them. We fully acknowledge that the relationship between flat minima and generalization remains an open and nuanced research question. Rather than taking a definitive stance in this ongoing debate, our work aims to contribute to this conversation by demonstrating that a regularization strategy informed by Fisher information and sharpness can lead to improved robustness and generalization in real-world time series tasks. Importantly, we have taken care to avoid overclaims in the paper, using qualified language such as "potential" and "achievable" to reflect the limitations inherent in this area. While authors in (Dinh et al., 2017a; Petzka et al., 2021) raise concerns about its limitation, these results are derived under specific assumptions (e.g., fully connected ReLU networks and carefully constructed reparameterizations). Their applicability to general architectures and practical training setups remains limited.

Moreover, recent empirical studies (Jiang et al., 2019; Andriushchenko & Flammarion, 2022) suggest that in practical settings, where such reparameterizations are not applied, sharpness (as commonly measured) can still correlate meaningfully with generalization. These observations support the idea that sharpness-based metrics, while theoretically imperfect, can still provide practical value. In addition, as discussed in our related work section, several recent papers (Neyshabur et al., 2017; Zhang & Xu, 2024; Foret et al., 2020; Andriushchenko & Flammarion, 2022; Kim et al., 2022a; Yun & Yang, 2024) supported the utility of sharpness-related methods and successfully leveraged them to improve learning outcomes.

Therefore, we believe our results add to this growing body of evidence, particularly in the underexplored domain of time series data, and we remain cautious yet optimistic about the promise of these methods.

## H. Graphic Summary

See Fig. S11.

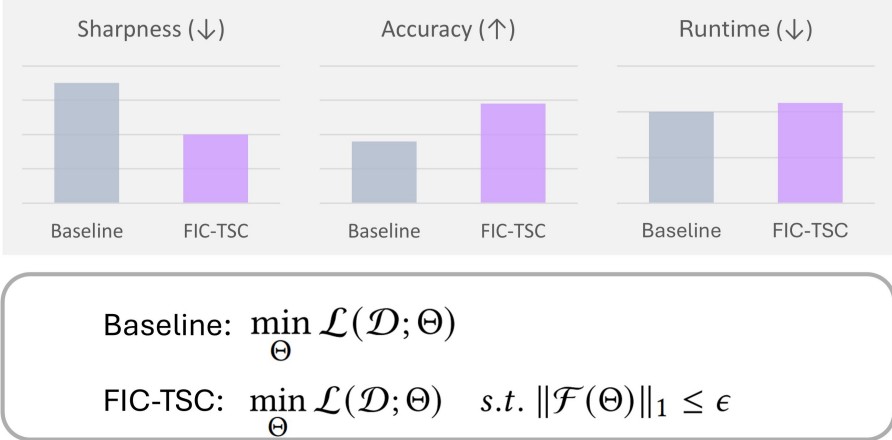

*Figure S11.* Comparison of the baseline method (standard training) and our proposed FIC-TSC approach. Training with FIC-TSC leads to convergence at a flatter minimum, potentially enhancing performance. The additional runtime incurred is insignificant and considered negligible.

