# OpenReview forum: "FIC-TSC: Learning Time Series Classification with Fisher Information Constraint"
_ICML.cc/2025/Conference — ICML 2025 poster_

### Official Review · Reviewer_YXQa · 2025-02-18

**Overall Recommendation:** 3

**Summary:**

The paper  introduces a novel framework for time series classification (TSC) that addresses domain shift issues by leveraging Fisher information as a constraint.

Main Contributions:
Domain Shift Problem in TSC: The paper highlights the challenge of domain shifts in time series classification, where the test set distribution deviates from the training set, leading to reduced classification accuracy. It examines the limitations of existing normalization-based solutions, such as Reversible Instance Normalization (RevIN), which were effective in regression tasks but ineffective in classification.

Fisher Information Constraint (FIC-TSC):
The proposed method incorporates Fisher Information Constraint (FIC) into training, which guides neural networks toward flatter minima, enhancing generalization under distribution shifts.
Direct computation of Fisher Information Matrix (FIM) is computationally expensive, so the authors use a diagonal approximation and gradient re-normalization to efficiently impose the Fisher information constraint.

Theoretical Insights:
The method is justified mathematically, linking Fisher information to sharpness-aware minimization and showing that networks trained with FIC achieve flatter minima. Theoretical convergence guarantees are maintained while improving generalization.

Empirical Validation:
The method is evaluated on 30 UEA multivariate and 85 UCR univariate time series classification datasets.
Results show superior performance over 14 state-of-the-art models, including TsLaNet, GPT4TS, ROCKET, and TimesNet.
The method is computationally efficient, achieving better generalization without requiring an additional backward pass per iteration.

Main Findings:
FIC-TSC outperforms prior approaches in handling domain shifts and achieves higher classification accuracy across diverse time series datasets.
The proposed Fisher Information Constraint reduces sharpness in loss landscapes, leading to better robustness and generalization.
The approach is efficient, requiring only a single backward pass per iteration, unlike prior sharpness-aware methods such as SAM.

Key Takeaways:
FIC-TSC introduces a novel constraint-based optimization approach that significantly improves time series classification under domain shifts.
It is both theoretically grounded and practically effective, making it a promising advancement in time series analysis.

**Claims And Evidence:**

1. Claim: “Domain shifts in TSC degrade classification performance.”
Evidence & Analysis:
The authors illustrate distribution discrepancies in the UEA datasets by plotting histograms for selected train/test sets and computing Wasserstein-1 distances.
They also show Reversible Instance Normalization (RevIN) helps reduce shift in regression but not in classification, motivating the need for an alternative method.
Verdict: These examples, while drawn from a subset of datasets, convincingly demonstrate that train–test divergence is common enough to undermine classification accuracy.
2. Claim: “Constraining Fisher information leads to flatter minima and improved generalization.”
Evidence & Analysis:
Theoretical Foundation: The authors use the known relationship between Fisher Information Matrix (FIM) and the Hessian at a local optimum, arguing that bounding Fisher information fosters lower sharpness.
Diagonal Approximation & Gradient Renormalization: To keep computation feasible, they approximate the FIM by its diagonal and rescale gradients if the FIM norm exceeds a threshold. They prove the approach retains an 𝑂(1/𝑇)O(1/T) convergence rate.
Empirical Sharpness Reduction: Post-training, they compare the sharpness of baseline vs. Fisher-constrained models, showing the latter achieves consistently flatter minima.
Verdict: Although off-diagonal elements are ignored, the approximation appears effective. Empirical results suggest that the constraint indeed decreases sharpness.
3. Claim: “FIC-TSC outperforms 14 state-of-the-art methods on 30 UEA and 85 UCR datasets.”
Evidence & Analysis:
The authors compare their approach (with both universal and dataset-specific hyperparameters) to a broad slate of methods, including ROCKET, InceptionTime, PatchTST, and TimesNet, showing average accuracy gains in the 1–3% range.
They incorporate Wilcoxon signed-rank tests to highlight the statistical significance of improvements.
Verdict: The large-scale experiments are thorough. Since these are standard splits, additional “real shift” setups (e.g., temporal splits) could further confirm the model’s shift-handling capability. However, the reported gains over many baselines look convincing.
4. Claim: “Our Fisher-based method is more efficient than other sharpness-aware solutions.”
Evidence & Analysis:
FIC-TSC uses only one backward pass per mini-batch, while methods like SAM require two, roughly doubling computation.
The authors compare iteration-level runtimes on selected datasets to demonstrate FIC-TSC’s speed advantage.
Verdict: Given standard deep-learning frameworks, a single-pass approach should indeed be faster. This is plausible and supported by iteration-time measurements.
Potential Weaknesses:
Diagonal Approximation
While standard in large-scale second-order methods, ignoring off-diagonal terms could limit accuracy of curvature estimates. An ablation or discussion on how robust performance is under this approximation might enhance confidence.
Realistic Domain Shift Splits
Most experiments still rely on standard train/test divisions from UCR/UEA. Inducing controlled or temporal shifts in the data could help illustrate exactly how robust FIC-TSC can be in true real-world drift scenarios.
Hyperparameter Sensitivity
The paper briefly explores 𝜖 settings (the Fisher constraint threshold) but does not deeply map how different ϵ values influence final accuracy or model stability across varied tasks.
Final Verdict
Overall, the paper’s key contributions—(i) identifying domain shift issues in TSC, (ii) introducing Fisher-based constraints to improve generalization via flatter minima, and (iii) empirically surpassing a range of strong baselines—are supported by both theoretical arguments and extensive benchmark results. While points like diagonal approximation, real-world shift tests, and hyperparameter exploration merit deeper discussion, the evidence is generally strong enough to validate the authors’ central claims.

**Essential References Not Discussed:**

1. Prior Work on Fisher Information / Second-Order Methods
Kronecker-Factored Approximate Curvature (K-FAC)
Reference: Martens & Grosse, ICML 2015, “Optimizing Neural Networks with Kronecker-Factored Approximate Curvature.”
Relevance: Demonstrates a practical way to approximate second-order information using Kronecker-factored matrices, thus improving training efficiency without resorting strictly to diagonal approximations. While K-FAC is mainly used for faster convergence, it also relates directly to the Fisher Information Matrix. The current paper’s diagonal-Fisher approach could be contrasted with or informed by this more elaborate approximation.
Elastic Weight Consolidation (EWC)
Reference: Kirkpatrick et al.  PNAS 2017, “Overcoming catastrophic forgetting in neural networks.”
Relevance: Uses Fisher Information to preserve previously learned knowledge when new tasks arrive. Even though EWC focuses on continual learning, its reliance on Fisher-based constraints parallels this paper’s method for controlling sharpness. Citing EWC would acknowledge foundational uses of Fisher Information in deep learning optimization and illustrate that the notion of stabilizing parameter updates via FIM is an active line of research.
2. Domain Adaptation / Distribution Shift in Time Series
Deep Domain Adaptation for Time Series
Example Reference: Purushotham et al., KDD 2017, “

**Ethical Review Concerns:**

Based on the content and scope of this paper—an algorithmic and methodological contribution focused on time series classification—I see no obvious ethical concerns regarding data misuse, participant harm, or similar issues. The datasets involved (UEA/UCR) are publicly available and widely used in research, and the paper does not suggest any problematic data collection or privacy violations. Thus, there is no apparent need to flag the paper for ethics review.

**Experimental Designs Or Analyses:**

1. Scope of Datasets and Baselines
UEA (30 datasets) and UCR (85 datasets):
These are highly regarded open-source repositories covering both univariate and multivariate TSC, offering a broad distribution of domains (e.g., sensor signals, ECG data, image outlines).
Validity: Using these archives is widely accepted for benchmarking TSC models, thus the selection is appropriate and representative.
Baseline Comparisons:
The paper benchmarks against 14 diverse methods, including both classic (e.g., ROCKET, InceptionTime) and modern (e.g., TimesNet, PatchTST) approaches.
Validity: This large set of baselines, covering a variety of algorithmic designs, strengthens the credibility of the results.
Statistical Tests:
They employ Wilcoxon signed-rank tests to compare model accuracies across multiple datasets.
Validity: The Wilcoxon test is a standard, non-parametric choice in multi-dataset settings. It bolsters confidence that reported gains are not merely coincidental or dataset-specific.
2. Experimental Protocols
Train/Test Splits:
They follow the standard UCR/UEA data partitions rather than generating alternative splits.
Potential Issue: These standard splits do not always strictly reflect real-world distribution shifts (e.g., chronological shifts). However, the authors do point out that some train–test differences already exist (shown via histograms/Wasserstein distances). More explicit or controlled shift scenarios might have provided stronger evidence of domain-robustness.
Metrics:
Accuracy is the primary metric, supplemented by F1, precision, and recall for selected experiments.
Validity: Accuracy is typical for classification; adding more metrics helps address class imbalance or multi-class comparisons. This is consistent with established TSC practices.
Hyperparameter Selection:
The authors discuss two strategies: a “universal” (same hyperparameters for all datasets) and a “full” (dataset-wise tuning).
Validity: Showing both strategies clarifies how stable the method is when hyperparameters are not exhaustively tuned, and how much improvement is possible with targeted tuning. This split is a fair approach and demonstrates practical applicability.
Computational Efficiency Reporting:
The paper compares runtime per iteration with Sharpness-Aware Minimization (SAM), illustrating that the proposed Fisher-based approach is lighter due to its single-pass design.
Validity: Presenting runtime differences is important for real-world feasibility. They show consistent speed gains over SAM, which seems credible given SAM’s two backward passes.
3. Analysis of Results
Average Accuracies:
The authors list comprehensive tables of accuracy scores for all 115 datasets, often showing 1–3% gains over competitive methods.
They use statistical significance tests (Wilcoxon) to confirm whether these gains are systematic.
Validity: Reporting means plus significance metrics is standard in large-scale TSC evaluations. The improvement margins appear reasonably consistent across many datasets.
Sharpness Measures:
They measure and visualize sharpness or curvature in the parameter space to show how the proposed method yields flatter minima.
Validity: These additional visual/quantitative analyses back up the theoretical claims. While each sharpness metric is an approximation, it does support their notion that the method reduces sensitivity to domain shifts.
Ablation Studies:
They vary 𝜖 (the Fisher norm threshold) to see how it affects performance.
Potential Issue: Some readers might want deeper exploration of the hyperparameter’s sensitivity across more datasets. The paper gives partial insights but could further detail the trade-offs between different 𝜖 values in a more systematic manner.
4. Overall Soundness
Positives:
Large benchmark coverage (115 datasets), multiple baselines, and recognized statistical tests.
Clear demonstration of both universal vs full hyperparameter settings, indicating robustness.
Reported run-time comparisons for efficiency claims.
Minor Caveats:
Standard train/test splits in UCR/UEA do not precisely mirror real-world domain shifts, although the authors do show partial evidence of distribution differences.
The hyperparameter search could be more comprehensively documented to fully assure reproducibility, though the current studies do give enough for a fair comparison.
Conclusion:
The paper’s experimental designs and analyses generally adhere to standard TSC practices and provide robust support for the proposed method. Although more targeted shift experiments or extended hyperparameter sensitivity tests could strengthen the domain robustness argument, the large-scale evaluations, statistical comparisons, and runtime analyses all appear fundamentally sound and valid.

**Methods And Evaluation Criteria:**

1. Proposed Methods
1.1 Fisher Information Constraint (FIC)
Core Idea:
The authors introduce a Fisher Information Constraint to encourage flatter minima. They approximate the Fisher Information Matrix (FIM) by its diagonal and rescale gradients if the overall Fisher norm surpasses a threshold 𝜖
Implementation:
Unlike double-backward techniques (e.g., SAM), FIC-TSC needs only a single backward pass each iteration.
This keeps overhead low while preserving benefits of second-order information.
Why It Makes Sense:
Time series data often exhibit domain shifts, where distributions differ between training and testing. FIC encourages the model to be less sensitive to small input perturbations, boosting robustness.
The diagonal approximation is a practical trade-off, capturing enough curvature to reduce sharpness without excessive computation.
1.2 Addressing Domain Shifts
Non-Stationarity:
Many real-world TSC tasks—sensor data, medical signals—can shift over time. A sharper minimum can lead to overfitting.
By bounding the Fisher norm, the paper aims to reduce such overfitting, achieving better generalization under unseen conditions.
2. Evaluation Criteria
2.1 Datasets
UEA (Multivariate) and UCR (Univariate)
Cover 30 and 85 datasets respectively, spanning a wide array of TSC problems.
They are standard archives in the TSC community.
While standard splits may not always replicate “hard” domain shifts, they do exhibit real differences between train/test distributions.
2.2 Metrics and Comparisons
Accuracy:
The paper primarily uses classification accuracy, supplemented by additional metrics like F1 scores on some datasets.
Statistical Tests:
Wilcoxon signed-rank tests assess significance of accuracy improvements.
Baselines:
Comparison with 14 methods, including ROCKET, InceptionTime, TimesNet, and PatchTST.
This broad coverage clarifies where FIC-TSC stands versus both older and newer TSC models.
2.3 Suitability
The authors’ approach aligns with typical TSC evaluation practices—large-scale experimentation on UCR/UEA is widely recognized as a benchmark standard.
Reporting both mean accuracy and statistical tests is methodologically sound, ensuring that improvements are not mere artifacts of specific datasets or chance.
3. Overall Alignment with the Problem
Focus on Distribution Shifts
TSC often suffers from training–testing mismatches, so a method that systematically curbs parameter sensitivity is well-justified.
Efficiency
FIC-TSC’s single backward pass makes it more practical than other sharpness-aware methods, especially for larger networks or frequent online training updates.
Comprehensive Benchmarking
Evaluating on 115 total datasets (30 UEA + 85 UCR) ensures results are not tied to one domain.
The authors also highlight relevant statistics (precision, recall, F1) and run significance tests, reinforcing the robustness of their comparisons.
Conclusion
FIC-TSC directly targets a core challenge in time series classification: the impact of domain shifts on model performance. By constraining the Fisher Information via a diagonal approximation, the method encourages learning a flatter solution that is less sensitive to distributional changes. The authors’ evaluation strategy—using well-known UCR/UEA benchmarks, reporting standard metrics, and conducting statistical significance tests—effectively demonstrates the method’s advantages over existing TSC models. Hence, the proposed methods and evaluation align well with the problem, showing both conceptual appropriateness (mitigating domain shifts) and empirical thoroughness (large-scale comparisons on standard archives).

**Other Comments Or Suggestions:**

Below are additional suggestions and minor observations:

Hyperparameter Tuning Explanation
It would be helpful to have a more detailed discussion in the text on how you searched or selected the threshold 𝜖 for the Fisher Information Constraint. Although the paper mentions a small grid search, providing explicit rationale or heuristics would benefit readers trying to replicate or extend the approach.
Results Section Organization
You might consider splitting the main performance tables into (a) univariate (UCR) and (b) multivariate (UEA) subtables for clarity. The current combined presentation is still understandable but separating them could help readers compare methods more intuitively by domain type.
Highlighting Real-World Case Studies
If any of the UCR/UEA datasets map closely to real industrial or medical shift scenarios, explicitly mentioning them might further strengthen the practical motivation. A short  example could illustrate why flatter minima matter in that setting.

**Other Strengths And Weaknesses:**

Strengths
Novel Combination of Existing Ideas

The paper blends known second-order optimization insights (i.e., Fisher Information, Hessian relationships) with a specific target of distribution shifts in time series classification. While Fisher-based methods are not new, applying them as a one-pass constraint to flatten minima for TSC is an original twist.
Significance and Broad Applicability
Domain shift problems are pervasive in real-world time series tasks (e.g., evolving sensor conditions, non-stationary signals). Demonstrating a general solution that can integrate with both convolution- and transformer-based TSC models enhances the paper’s practical value.
Clarity in Writing and Structure
The paper is generally well-structured. The motivation (domain shift in TSC), main idea (constrain Fisher norm), and experimental evaluation (comprehensive benchmarking on UCR/UEA) are laid out in a logical progression. Key points—like the contrast to Reversible Instance Normalization—are relatively clear and straightforward to follow.
Extensive Experiments and Statistical Significance
The authors present results on 115 datasets, with multiple baselines, and use Wilcoxon signed-rank tests. This thorough coverage supports the claim that the method is widely applicable and not narrowly tuned to a handful of problems.
Weaknesses
Limited Exploration of Realistic Shift Scenarios
Although the paper shows distribution differences in standard train/test splits, it does not deeply investigate scenarios like chronologically separated training and testing. A controlled shift experiment (e.g., training on earlier time frames, testing on later frames) could further validate real-world domain-robustness.
Diagonal Approximation Restricts Full Second-Order Information
While efficient, ignoring off-diagonal terms may limit capturing interactions among parameters. An ablation comparing diagonal vs. block-diagonal approximations (or referencing Kronecker-factored approaches) would strengthen the paper’s second-order argument.
Hyperparameter Sensitivity Analyses
The paper briefly discusses setting the threshold ϵ but does not provide a thorough grid-based sensitivity exploration across multiple datasets. More systematic experimentation could clarify how much
ϵ-tuning is required for robust performance.
Somewhat Limited Theoretical Details
While the authors offer a high-level sketch of convergence arguments, more detailed proofs or bridging steps could help. In particular, it would be helpful to see how quickly the algorithm converges in practice under typical TSC conditions (e.g., smaller sample sizes, higher dimensional signals).
Overall Assessment
Originality: The idea of enforcing Fisher constraints in a single-pass framework, specifically tailored to time series classification, stands out for handling domain shifts—an underserved area in TSC compared to the often-solved architecture innovations.
Significance: Given that domain shifts are a prevalent real-world challenge, the approach has practical and theoretical importance.
Clarity: The paper is mostly coherent and well-motivated, though a deeper discussion of certain hyperparameters or more elaborate shift scenarios might enhance understanding.
Despite some noted weaknesses, the paper’s strengths—innovative re-interpretation of Fisher-based constraints for TSC, extensive comparative experiments, and strong performance—indicate a valuable contribution to time series research.

**Questions For Authors:**

Below are several questions that address points where additional clarification or detail could potentially change the evaluation of the paper:

Explicit Domain Shift Scenario

Question: Have you conducted any experiments using a chronologically split setup or another explicit shift scenario (e.g., training on earlier data, testing on later data) to confirm the method’s robustness beyond standard UCR/UEA splits?
Why It Matters: If FIC-TSC demonstrates notable improvements under controlled real-world shifts, that would further validate the claims about domain-shift resilience. Conversely, if such tests haven’t been done, it’s possible that the standard splits do not fully capture how well the method handles non-stationary data.
Hyperparameter Sensitivity

Question: Please provide more systematic results on the impact of different ϵ values across multiple datasets, and explain how a practitioner might choose ϵ for new tasks?
Why It Matters: The threshold ϵ appears central to the Fisher constraint’s effectiveness. Detailed guidance, or an ablation over a broader range of datasets, would clarify how sensitive the method is to this parameter and whether it needs per-dataset tuning.
Diagonal Approximation Trade-Off
Question: Have you tested any partial or block-based FIM approximations to see if ignoring off-diagonal terms significantly affects performance or memory demands?
Why It Matters: This would reveal whether the diagonal approximation is sufficiently capturing curvature or if a more nuanced approximation could yield further improvements (albeit at higher computational cost).
Comparison to Domain Adaptation Methods
Question: How does FIC-TSC compare with existing domain adaptation or transfer learning approaches for time series, especially methods that explicitly align distributions (e.g., adversarial alignment techniques)?
Why It Matters: Although your focus is on robust optimization, the domain-adaptation community also tackles shifts in time series. Clarifying these potential synergies or differences may strengthen the positioning of FIC-TSC in the broader literature.
Runtime and Memory Scaling
Question: Beyond single-batch iteration timing, did the authors benchmark FIC-TSC’s training speed and memory use on large-scale datasets (e.g. tens of thousands of training samples) or with very deep networks?
Why It Matters: Demonstrating consistent scaling properties can solidify the paper’s claim that FIC-TSC is more efficient than other sharpness-aware approaches (SAM) in practical, large-scale scenarios.

**Relation To Broader Scientific Literature:**

1. Fisher Information and Sharpness-Aware Optimization
Prior Work:
Sharpness-aware methods like SAM (Sharpness-Aware Minimization) introduce a second backward pass to encourage flatter minima. In parallel, Fisher Information has long been connected to curvature in information geometry, relating to the Hessian of log-likelihoods.
Paper’s Advancement:
This paper combines these strands by using Fisher information as a direct constraint, making the optimization process single-pass rather than double.
It thus situates itself alongside second-order and sharpness-focused approaches but does so with a diagonal FIM approximation, keeping computational overhead minimal.
2. Time Series Classification under Domain Shifts
Prior Work:
TSC literature often centers on specialized architectures (InceptionTime, ROCKET, various Transformers) or similarity-based methods (DTW). However, handling non-stationarity or train–test distribution shifts in TSC has received comparatively less attention beyond methods like Reversible Instance Normalization (RevIN), which primarily aids forecasting/regression tasks.
Paper’s Advancement:
By directly addressing distribution shifts within TSC (rather than purely focusing on new architectures), the paper provides a relatively novel perspective: it treats domain shifts as a fundamental optimization/robustness issue, not merely a data-augmentation or normalization challenge.
This complements prior TSC work on normalization techniques by proposing that regularizing curvature (via Fisher Information) can outperform, or at least fill the gap left by, normalization-based solutions in classification settings.
3. Empirical Benchmarks and Method Comparisons
Prior Work:
Many TSC papers rely on UCR/UEA archives for evaluation, but most emphasize raw accuracy improvements with domain-specific innovations (CNN designs, ensemble methods, or transform-based approaches). Ensemble methods like HC2 or shapelet-based frameworks have historically been robust but often with higher computational cost.
Paper’s Advancement:
The authors show that a general optimization-level technique—FIC—can match or surpass specialized TSC architectures on the same standard benchmarks.
This demonstrates that certain general-purpose enhancements (aimed at stability or curvature control) can be as crucial as sophisticated network designs in pushing performance bounds.
4. Connection to Information Geometry and Generalization
Prior Work:
Fisher Information has been used to study generalization, but mostly in contexts such as Bayesian inference, elastic weight consolidation, or advanced network compressions. These approaches often require extra memory or partial second-order updates.
Paper’s Advancement:
By framing Fisher Information as a constraint on gradient norms (rather than storing or inverting any matrix blocks), the paper contributes a simpler, more scalable variant of these second-order ideas.
It thereby extends the principle that controlling curvature (through the FIM) can guard against overfitting, especially in domains—like TSC—where distribution shifts are common.
5. Overall Positioning
Bringing Second-Order Insights to TSC:

The paper merges second-order optimization insights with TSC challenges, emphasizing domain shift resilience rather than purely augmenting classification architectures.
Bridging Normalization Gaps:

Techniques like batch or instance normalization have been central in other time series tasks (especially forecasting). The paper positions itself as a complementary or alternative approach for classification scenarios where direct normalization sometimes obscures class differences.
Potential for Future Extensions:
This approach could be extended or integrated with other robust classification methods (e.g., adversarial training, domain adaptation), indicating synergy with broader machine learning methods beyond TSC.

Conclusion:
In summary, the paper’s key contributions—a Fisher-constraint–based approach for flatter minima, tailored specifically to TSC’s domain shift challenge—fit well into existing second-order/sharpness-aware frameworks while tackling a recognized but less-explored problem in the TSC literature. It bridges general-purpose optimization insights (FIM-based regularization) with domain-specific concerns (non-stationarity in time series), making it a noteworthy addition to both communities.

**Theoretical Claims:**

1. Overview of Theoretical Claims
Equivalence of Fisher Information and Hessian

The paper states that the Fisher Information Matrix (FIM) is asymptotically equivalent to the Hessian of the negative log-likelihood at a local optimum.
This is a well-established result in information geometry. The authors’ statement aligns with known derivations (e.g., under regularity conditions, the expected Hessian and FIM coincide).
Sharpness Reduction via Fisher Constraint

They link Fisher-based constraints to reduced sharpness, arguing that smaller FIM norms imply flatter minima.
The proof involves approximating sharpness around a local minimum using a Taylor expansion and relating the Hessian to the FIM. The steps appear consistent with prior sharpness-aware minimization work.
Convergence Rate

The paper claims that imposing the Fisher constraint preserves an O(1/T) convergence rate under standard smoothness assumptions.
The authors outline a gradient-based convergence analysis, treating the constraint as a re-normalization step. While not exhaustive, the argument follows typical first-order proof templates and does not introduce obvious contradictions.
2. Checked Details and Issues
FIM–Hessian Equivalence:
The authors’ derivation is standard, relying on well-known results in maximum likelihood theory. No major oversights were found.
Flatness and Taylor Expansion:
The paper’s link between smaller diagonal FIM and flatter minima uses the logic that the Hessian’s diagonal dominates local curvature in a diagonal approximation. Though it omits cross-terms, the rationale is mathematically sound for large-scale deep networks that rely on diagonal approximations.
One potential limitation is that ignoring off-diagonal terms may underestimate curvature in certain directions, but this is disclosed as a simplifying assumption rather than a full second-order analysis.
Convergence Proof:
The authors provide a high-level sketch showing that re-normalizing gradients when the FIM norm exceeds a threshold does not break standard Lipschitz-based convergence arguments.
A full, line-by-line formal proof with all constants and step sizes is not fully detailed, but the outline is in line with existing gradient-descent proofs.
3. Conclusion
Overall, the theoretical claims appear largely correct and adhere to established principles in optimization and information geometry:
FIM–Hessian Relationship: Properly stated and well-known in literature.
Sharpness Reduction: Reasonably extended from the FIM–Hessian link and standard Taylor expansions.
Convergence Rate: Based on recognized gradient-descent proofs, with a plausible argument that enforcing an upper bound on Fisher norm does not slow the asymptotic 𝑂(1/𝑇) rate.
While the diagonal approximation and partial discussion of cross-terms mean the proofs do not capture every nuance of a full second-order method, they are consistent with standard practices and do not exhibit fundamental errors.

---

> ### Author Rebuttal · Authors · 2025-03-29
>
> We deeply appreciate the reviewer's effort in carefully reviewing our paper and giving very constructive suggestions. We also thank the reviewer for recognizing our novelty, significance, and theoretical and empirical analysis.
>
> ---
>
> ### **Explicit Domain Shift Scenario.**
>
>  We consider the following two scenarios.
> - **Healthcare dataset**: Splitting data by patient introduces explicit domain shift due to physiological variability. We use four popular public datasets (anonymized in accordance with ethical standards): TDBrain, ADFTD, PTB-XL, and SleepEDF.
>
> - **Online Handwriting Recognition (OnHW-Chars)**: The training set contains characters from **right-handed** writers, while the test set features **left-handed** writers, who often differ in stroke direction, pressure, slant, and orientation. Additionally, the two datasets were released at different times, introducing potential temporal shifts.
>
> **Our method outperforms the baseline with accuracy gains of 1.6%–6.8% and F1 improvements of 0.7%–4.3%.**
>
> **View full results at https://github.com/AnonymousUserss/ICML2025-4119-Response.**
>
>
> |Dataset|Method|Acc.|
> |-|-|-|
> |TDBrain|Baseline|93.0|
> ||**+FIC**|**96.2**|
> |ADFTD|Baseline|46.0|
> ||**+FIC**|**52.8**|
> |PTB-XL|Baseline|73.9|
> ||**+FIC**|**75.1**|
> |SleepEDF|Baseline|85.1|
> ||**+FIC**|**86.7**|
> |OnHW-Char (R→L)|Baseline|44.1|
> ||**+FIC**|**46.8**|
>
>
> [1] Wang, Yihe, et al. "How to evaluate your medical time series classification?.".
>
> ---
>
> ### **Systematic Results on $\epsilon$.**
>
> Please refer to our response to **Reviewer kJBN Q2**.
>
> ---
>
> ### **Diagonal Approximation Trade-Off Question.**
>
> Yes, we did. Computing the full Fisher Information Matrix is theoretically and practically expensive, substantially growing the time complexity from $O(n)$ to $O(n^2)$.
> As noted in Appendix G.4, our empirical analysis confirms that diagonal approximation is essential for practical use. **We tested the computation of the full FIM using a mini-batch size of 64 in a single iteration of InceptionTime (about 0.6M parameters) on an NVIDIA A100 40GB GPU. This operation took over 15 minutes per iteration, rendering full FIM computation infeasible for real-world training.** In contrast, with the diagonal approximation, the same computation can be completed in approximately 0.1 seconds (see Table 5), demonstrating a significant improvement in efficiency.
>
> ---
>
> ### **Domain Adaptation/Transfer Learning.**
>
> Domain adaptation and transfer learning address distribution shifts by enabling models trained on a source domain to generalize to a related target domain. Common strategies include aligning feature distributions by minimizing statistical distances [2] or using adversarial training to make features indistinguishable across domains [3,4].
>
> In contrast, our method is orthogonal to these approaches. It does not require access to target domain data or labels during training, making it suitable when the target distribution is unknown or unavailable.
> **Importantly, domain adaptation/transfer learning could be applied as a post-training or downstream enhancement once target domain data becomes available.** In this sense, our method and these techniques can be complementary.
>
> [2] HoMM: Higher-order moment matching for unsupervised domain adaptation AAAI 2020
>
> [3] Purushotham, Sanjay, et al. "Variational recurrent adversarial deep domain adaptation." ICLR 2017.
>
> [4] Jin, Xiaoyong, et al. "Domain adaptation for time series forecasting via attention sharing." International Conference on Machine Learning. PMLR, 2022.
>
>
> ---
>
> ### **Runtime and Memory Scaling Question.**
>
> We scale PatchTST by setting different numbers of blocks. The test is conducted on InsectWingbeat (30k training samples). **Our method reduces memory usage by 5–8% and cuts runtime by more than 50%.**
>  |Model|#Params|Method|Memory(MB)|Time/Epoch(s)|
> |:-:|:-:|:-:|:-:|:-:|
> |PatchTST-1|4.5M|Ours|**706**|**1.8**|
> |||SAM|768|4.3|
> |PatchTST-5|17.1M|Ours|**1022**|**5.2**|
> |||SAM|1106|12.3|
> |PatchTST-20|64.4M|Ours|**2254**|**18.4**|
> |||SAM|2370|43.3|
>
> ---
>
> ### **Essential References to Be Included.**
>
> EWC preserves prior knowledge by penalizing changes to important weights, using a Gaussian posterior centered at previous weights with precision from the observed Fisher information (Laplace approximation). Notably, **EWC uses a diagonal approximation, aligning with and supporting the efficiency goals of our work.**
>
> K-FAC addresses the high computational cost of the FIM by approximating large blocks of it, corresponding to entire layers, as the Kronecker product of two much smaller matrices. We consider this a promising direction for future work to achieve more accurate and efficient FIM approximations.
>
> [5] Overcoming catastrophic forgetting in neural networks
>
> [6] Optimizing Neural Networks with Kronecker-Factored Approximate Curvature
>
> ---
>
> **We will include the above discussion in our final version.**

---

### Official Review · Reviewer_QtfR · 2025-02-28

**Overall Recommendation:** 3

**Summary:**

FIC-TSC introduces a novel training framework for time series classification by enforcing a Fisher information constraint to guide the optimizer toward flatter minima, aiming to improve robustness against domain shift. The method leverages two key approximations—a diagonalized Fisher information matrix and a gradient normalization strategy that requires only one backward pass—to achieve computational efficiency. Experiments on standard UEA and UCR datasets demonstrate that FIC-TSC outperforms several state-of-the-art methods in terms of classification accuracy and runtime, while also reducing the sharpness of the loss landscape.

## Update After Rebuttal
I raised my score as a result of a discussion with the authors (see below).

**Claims And Evidence:**

### Unjustified motivation.
The primary motivation of the paper is the assumption that "a flat minimum is less sensitive to small perturbations of parameters, and hence, is more robust to domain shifts" (p.2), as conceptually illustrated in Fig. 1. However, this assumption is not supported by rigorous theoretical or experimental evidence. The authors do not cite prior work that empirically or theoretically validates this assumption. Moreover, while Fig. 9 compares the loss landscapes of the baseline and the proposed method, both appear fairly flat, and there is no clear demonstration of how domain shifts would transform these landscapes.

### Questionable assumption.
The assumption above tells that if domain shift occurs, the loss landscape plotted in 2-D plane, where x- and y- axes are neural network weight parameters and loss value respectively, "slides" without changing its shape (Fig. 1). However, it is unclear why distribution shift in *data domain* leads to the slide in *weight domain*. As I mentioned above, this nontrivial hypothesis is neither verified theoretically or experimentally.

### Sharpness in data domain is not considered.
The paper’s analysis and visualizations (e.g., Fig. 1) are confined to the loss landscape in the weight parameter space. It implicitly assumes that under a domain shift, the loss landscape "slides" without changing its shape—a nontrivial claim that connects a shift in the data distribution to a simple translation in the weight domain. This assumption is neither theoretically justified nor empirically verified. Moreover, since domain shifts occur in the data domain, for the hypothesis that flat minima yield better cross-domain generalization to hold, the authors need to demonstrate that:
(i) existing approaches produce sharp minima when visualized in the data domain,
(ii) FIC-TSC results in flat minima in the data domain, and
(iii) FIC-TSC outperforms baselines in cross-domain experiments. Without such evidence, the connection between parameter-space flatness and robustness to domain shifts remains unproven.

### Questionable link between flatness and generalization.
The relationship between flat minima and generalization is still a matter of debate. Works such as Dinh et al. (2017) ("Sharp Minima Can Generalize For Deep Nets") have shown that conventional flatness or sharpness measures are not invariant under reparameterizations—meaning that a flat minimum can be transformed into an arbitrarily sharp one without changing the network’s function. Similarly, Petzka et al. (2021) ("Relative Flatness and Generalization") argue that generalization is more closely tied to feature robustness than to absolute flatness in parameter space. The failure to discuss these critical findings raises concerns about the validity of the paper’s central premise.

### Concern on reproducibility.
The experimental results are reported as scalar values without error bars or confidence intervals, making it challenging to evaluate the stability and robustness of the proposed approach. This lack of statistical rigor raises doubts about the significance of the observed performance gains. Furthermore, without a public release of the code, it is even more difficult for reviewers and future researchers to assess the reproducibility and practical limitations of the method.

### Technical simplifications are not adequately evaluated.
FIC-TSC introduces two technical simplifications for computational efficiency: (1) the diagonal approximation of the Fisher information matrix, and (2) enforcing the Fisher constraint using a single backward pass rather than a costly double back-propagation. While these tricks reduce computational load, the paper does not provide a rigorous ablation study comparing FIC-TSC with a variant that uses the full Fisher matrix and/or double back-propagation—even on simple artificial datasets or small networks. Without such controlled experiments, it is difficult to assess whether these approximations negatively impact the method’s robustness or generalization performance.

**Essential References Not Discussed:**

Please see the above Claims And Evidence section.

**Experimental Designs Or Analyses:**

Please see the above Claims And Evidence section.

**Methods And Evaluation Criteria:**

Please see the above Claims And Evidence section.

**Other Comments Or Suggestions:**

The paper was an enjoyable read. Please note that my comments represent my initial impressions and may include misunderstandings. I welcome further discussion on these points and am open to revising my score once my questions and concerns are adequately addressed.

(Thank you for further clarification during the discussion period, I raised my score accordingly.)

**Other Strengths And Weaknesses:**

FIC-TSC's main strength lies in its computational efficiency—it leverages a diagonal Fisher information approximation and a single backward pass to enforce its constraint, making it potentially scalable and applicable to real-world time series classification scenarios.

**Questions For Authors:**

I have incorporated my questions into Claims And Evidence section.

**Relation To Broader Scientific Literature:**

The paper builds on a long-standing debate in the literature regarding the relationship between loss landscape flatness and generalization. Early works (e.g., Hochreiter & Schmidhuber, 1997; Keskar et al., 2017) linked flat minima to improved generalization, but subsequent studies (e.g., Dinh et al., 2017; Petzka et al., 2021) have shown that conventional flatness measures can be manipulated through reparameterizations, challenging their direct connection to generalization. FIC-TSC contributes by proposing a computationally efficient Fisher information constraint to enforce flatness, aiming to enhance robustness to domain shifts in time series classification, and thereby adds to the ongoing discussion by attempting to operationalize flatness in a practical setting.

**Theoretical Claims:**

I have reviewed the high-level theoretical claims but there remains a possibility that oversights exist.

---

> ### Author Rebuttal · Authors · 2025-03-29
>
> Thanks for these insightful suggestions, and we are very glad to hear that you enjoyed the reading.
>
> ---
>
> ### **Unjustified motivation/Questionable assumption/Sharpness in data domain.**
> - Our primary motivation is that time series data often suffers from domain shift between train and test set, and we propose to constrain FIM during learning to alleviate the issue, which potentially results in a flatter minimal and improves the generalization.
> - Regarding the concern about citing prior work for the claim that "a flat minimum is less sensitive ...", we have referenced studies earlier in the sentence (Keskar et al., 2016; Neyshabur et al., 2017; Zhang & Xu, 2024). Notably, in Keskar et al. (2016), the authors explicitly define sharpness as a measure of sensitivity (see p.5, Section 2.2.2). We will cite them in a more proper place in the final version.
> - **Fig. 1** is a conceptual illustration of landscapes with domain shifts. **We do not have the assumption that the shapes of the landscapes on test and train data remain the same.**
> - **The landscape is related to both data domain $\mathcal{D}$ and the weight domain $\Theta$, i.e., calculating loss $\mathcal{L}(\mathcal{D};\Theta)$ needs dataset and weights of a network**, so when training data and test data have domain shift, the landscapes with same weights can be different, i.e. $\mathcal{L}(D_{train};\Theta)$ and $\mathcal{L}(\mathcal{D}_{test};\Theta)$.
> **In Fig. 1, we use different colors to denote different data domains $\mathcal{D}$**. We improve it at https://github.com/AnonymousUserss/ICML2025-4119-Response.
> - **Fig. 9 is a visualization of the landscapes to better illustrate the concept and show the relative flatness reduction.** The **quantitative results** are presented in **Fig. 8** (not Fig. 9) and **Fig.5**, as the key evidence to support our claim: **Compared with baseline, using our method can obtain an average 40% reduction in the sharpness (see (i) and (ii)) across all datasets and finally translate to ~4% accuracy gain (iii).** Please refer to our response to **R#YXQa** for explicit domain-shift experiments.
>
> ---
> ### **Flatness and Generalization.**
> We thank the reviewer for highlighting these important works. We fully acknowledge that the relationship between flat minima and generalization remains an open and nuanced research question. **Rather than taking a definitive stance in this ongoing debate, our work aims to contribute to this conversation by demonstrating that a regularization strategy informed by Fisher information and sharpness can lead to improved robustness and generalization in real-world time series tasks.**
>
> **Importantly, we have taken care to avoid overclaims in the paper, using qualified language such as "potential" and "achievable" to reflect the limitations inherent in this area.**
>
> While Dinh et al. (2017) and Petzka et al. (2021) raise concerns about its limitation, **these results are derived under specific assumptions** (e.g., fully connected ReLU networks and carefully constructed reparameterizations). Their applicability to general architectures and practical training setups remains limited.
>
> Moreover, recent empirical studies [1-2] suggest that in practical settings, where such reparameterizations are not applied, **sharpness (as commonly measured) can still correlate meaningfully with generalization**. These observations support the idea that sharpness-based metrics, while theoretically imperfect, can still provide **practical value**.
>
> In addition, as discussed in our related work section, several recent papers (Zhang & Xu, 2024; Foret et al., 2020; Andriushchenko & Flammarion, 2022; Kim et al., 2022a; Yun & Yang, 2024; Ilbert et al., 2024) have
> - supported the utility of sharpness-related method;
> - successfully leveraged it to improve learning outcomes.
>
> **We believe our results add to this growing body of evidence, particularly in the underexplored domain of time series data, and we remain cautious yet optimistic about the promise of these methods.**
>
> [1] Fantastic generalization measures and where to find them
>
> [2] Towards Understanding Sharpness-Aware Minimization, ICML2022
>
> ---
>
> ### **Reproducibility.**
> **We have made the code and the trained weights at https://github.com/AnonymousUserss/ICML2025-4119-Response.**
>
> - **standard deviation (%)** on two benchmarks.
> ||Acc.|Bal. Acc|F1|P|R|
> |:-:|:-:|:-:|:-:|:-:|:-:|
> |UEA 30|1.1|1.2|1.4|1.5|1.2|
> |UCR 85 |0.5|0.8|0.9|1.4|0.8|
>
> ---
>
> ### **Technical Simplifications.**
> - Please see our response to **Reviewer 3cHV Q4** regarding diagonal approximation.
> - Double-backward vs single-backward with FIC is presented as follows. **Our method is on par with double-backward in accuracy but substantially reduces the runtime. Full comparison is at the same repo**.
>
> |Metrics|Accuracy(%)| |Runtime(s)| |
> |:-:|:-:|:-:|:-:|:-:|
> |10 datasets|Double|Ours|Double|Ours|
> |**Avg.**|77.4|76.8|0.147|0.070|

---

> > ### Comment · Reviewer_QtfR · 2025-04-03
> >
> > Thank you for your detailed and thoughtful response. I appreciate the hard work you put into addressing my concerns, especially your efforts in ensuring reproducibility by making your code publicly available and by providing standard deviation metrics for key benchmarks.
> >
> > However, my primary concern remains regarding the "sharpness in the data domain." While your explanation that the loss landscape $\mathcal{L}(\mathcal{D};\Theta)$ depends on both the data domain $\mathcal{D}$ and the weight space $\Theta$ is conceptually sound, the rebuttal does not offer explicit quantitative or visual evidence comparing the sharpness of the loss landscapes computed on training data versus those on shifted (test) data. My concern specifically pertains to the data-loss plane, which is critical for understanding the model's robustness to domain shifts, rather than the weight-loss plane, which is the primary focus of your current analysis.
> >
> > Given the importance of this issue to your central claims, I must maintain my original score until further evidence is provided that directly addresses the sharpness in the data domain.

---

> > > ### Author Response · Authors · 2025-04-04
> > >
> > > ## Thank you for taking the time to review our response. We appreciate your recognition of our efforts. Below, we address the remaining concerns.
> > >
> > > ---
> > >
> > > **Since sharpness values across datasets can differ in scale, we report the sharpness reduction, $reduction = \frac{Baseline- FIC}{Baseline}$.**
> > >
> > >
> > > ### Sharpness Reduction in Weight-Loss Plane
> > >
> > > We first show that our method can reduce sharpness (w.r.t weight) in both the training domain and the test domain. (We originally only showed a sharpness reduction in training data; as suggested, we added sharpness reduction on test data here).
> > >
> > >
> > > |dataset|EC|FD|HW|HB|JV|SCP1|SCP2|SAD|UW|PS|Avg.|
> > > |---|---|---|---|---|---|---|---|---|---|---|---|
> > > |Weight Sharp. Reduction on Training|0.31|0.12|0.45|0.13|0.82|0.10|0.44|0.84|0.32|0.34|0.41|
> > > |Weight Sharp. Reduction on Testing|0.11|0.10|0.22|0.31|0.24|0.73|0.34|0.31|0.16|0.33|0.29|
> > >
> > >
> > > As shown in the table, **our method can reduce sharpness by 41% and 29% w.r.t weight on training and test data, respectively**.
> > >
> > > ---
> > >
> > > ### Sharpness Reduction in Data-Loss Plane
> > >
> > > Similar to Sharpness defined on weight domain, the **sharpness on data domain**, at data $x$ with weight $\Theta$ is defined as
> > > $\text{Sharpness}(x) = \max_{x' \in \mathcal{B}_2 (\rho, x)}\frac{L(x', y;\Theta) - L(x, y;\Theta)}{1 + L(x, y;\Theta)}$,
> > > which measures the sensitivity of loss within a local neighborhood of input data, i.e. data sensitivity. Here, $\mathcal{B}_2 (\rho,x)$ is a Euclidean
> > > ball with radius $\rho$ centered at $x$. We present the data sharpness reduction on both the training set and the test below.
> > >
> > >
> > >
> > > |dataset|EC|FD|HW|HB|JV|SCP1|SCP2|SAD|UW|PS|Avg.|
> > > |---|---|---|---|---|---|---|---|---|---|---|---|
> > > |Data Sharp. Reduction on Training|0.42|0.59|0.89|0.72|0.87|0.63|0.88|0.54|0.49|0.37|0.64|
> > > |Data Sharp. Reduction on Testing|0.22|0.40|0.01|0.67|0.15|0.61|0.83|0.30|0.10|0.36|0.37|
> > >
> > >
> > > Similarly, **our method can reduce sharpness by 64% and 37% w.r.t data on training and test data, respectively.**
> > >
> > > ---
> > >
> > > ### Landscape Mismatch: Quantifying Generalization Gap
> > >
> > >  To better understand the reason for flatter minimal help generalization, we define a metric **landscape mismatch** to measure how **the difference between the training landscape and test landscape** at the local minimal $\Theta$:  $\Delta M = \int_{\Theta^{\prime}\in \mathcal{B}_2(\alpha, \Theta) } |L(D_test;\Theta^{\prime}) - L(D_train;\Theta^{\prime})| \text{d} \Theta^{\prime}$.
> > >
> > > Here, again $\mathcal{B}_2(\alpha, \Theta)$, a Euclidean ball with radius $\alpha$ centered at $\Theta$, and we use the Monte Carlo method to approximate $\Delta M$.
> > >
> > >
> > > The results are presented as follows. Again the reduction is calculated as $reduction = \frac{Baseline- FIC}{Baseline}$.
> > > |dataset|EC|FD|HW|HB|JV|SCP1|SCP2|SAD|UW|PS|Avg.|
> > > |---|---|---|---|---|---|---|---|---|---|---|---|
> > > |Train-Test Mismatch Reduction|0.42|0.08|0.19|0.23|0.19|0.28|0.47|0.36|0.16|0.41|0.28|
> > >
> > > The results suggest that **Our method can reduce the mismatch of landscape between test and training datasets** by 28%.
> > >
> > > ---
> > >
> > > In summary, these new analyses demonstrate that our method:
> > > - Reduces sharpness in both weight and data domains;
> > > - Achieves this reduction on both training and test data;
> > > - Mitigates the landscape mismatch between training and test data.
> > >
> > > Together, these results support the claim that our method encourages the model to converge to a flatter minimum through FIC, thereby translating this flatness into improved generalization performance.
> > >
> > > ---
> > >
> > > ## We hope our response addresses your concern. If you have any further questions, please let us know. Thanks!

---

### Official Review · Reviewer_kJBN · 2025-03-06

**Overall Recommendation:** 3

**Summary:**

This paper addresses the failure of RevIN in out-of-distribution (OOD) scenarios for time series classification and proposes a constraint method based on the Fisher Information Matrix to enable smoother model optimization, thereby improving the generalization ability of the classification model. Furthermore, considering the quadratic complexity of computing the Fisher Information Matrix and the need for two rounds of backpropagation, this paper introduces a diagonal approximation information matrix and a Fisher information constraint method to reduce the computational cost. The effectiveness of the proposed method is validated through performance evaluation experiments on the UEA 30 and UCR 85 datasets, case studies, sharpness evaluation, and landscape visualization.

## update after rebuttal

The authors' responses addressed my concerns, and I chose to maintain my original score.

**Claims And Evidence:**

Yes, the claims are convincing.

**Essential References Not Discussed:**

No.

**Experimental Designs Or Analyses:**

Yes, I have checked.

**Methods And Evaluation Criteria:**

Yes, the evaluation criteria and benchmark datasets are appropriate for the problem and application at hand.

**Other Comments Or Suggestions:**

No other comments or suggestions.

**Other Strengths And Weaknesses:**

Strengths:

1.	This paper clearly articulates the limitations of RevIN in out-of-distribution (OOD) scenarios for time series classification and innovatively leverages Fisher information constraints to mitigate OOD issues in time series classification.

2.	To address computational cost concerns, the paper proposes a diagonal approximation of the Fisher Information Matrix and a Fisher information constraint method.

3.	The experiments are comprehensive, including performance evaluation on UEA30 and UCR128 datasets, case studies, sharpness evaluation, and landscape visualization, which thoroughly validate the effectiveness of the proposed method.

Weaknesses:

1.	Although the authors suggest that using Fisher information can smooth the model's optimization space, the paper lacks a thorough analysis of whether and why time series data itself or the model inherently causes sharpness phenomena in time series classification.

2.	There is insufficient justification for the rationality of the diagonal approximation of the Fisher Information Matrix and the Fisher information constraint method.

**Questions For Authors:**

1.	It would be better to clarify why time series data lead to sharpness, improving the rationale and persuasiveness of the proposed method.

2.	Considering that Epsilon is a crucial hyperparameter, please conduct a sensitivity analysis and discussion of Epsilon.

3.	Compare with other advanced time series normalization methods, such as SAN [R1], Numerically Multi-scaled Embedding in NuTime [R2].

4.	Explain the rationality of the diagonal approximation of the Fisher Information Matrix under the premise of significant parameter correlation or strong model non-linearity (which is common in deep learning).

5.	Provide a reasonable explanation of how the Fisher information constraint reduces the Fisher Information Matrix.

6.	The authors state in the experimental section: "To fully explore the ability of our method, we perform a grid search for hyperparameters for each dataset." However, for the UCR 85 archive and the UEA 30 archive, each time series dataset only contains a training set and a test set, without a validation set. In the absence of a validation set, how did the authors conduct grid search to select hyperparameters?

[R1] Liu Z, Cheng M, Li Z, et al. Adaptive normalization for non-stationary time series forecasting: A temporal slice perspective. Advances in Neural Information Processing Systems, 2023, 36: 14273-14292.

[R2] Lin, Chenguo, et al. "NuTime: Numerically Multi-Scaled Embedding for Large-Scale Time-Series Pretraining." Transactions on Machine Learning Research, 2024.

**Relation To Broader Scientific Literature:**

No.

**Theoretical Claims:**

Yes, I have checked.

---

> ### Author Rebuttal · Authors · 2025-03-29
>
> We sincerely thank the reviewer for the thoughtful and insightful comments, especially for recognizing our contributions and empirical validation. We addressed the main concerns as follows.
>
> ---
>
> ### **Q1. Data and Sharpness.**
>
> Time series datasets are often small in size (e.g., UW has 120 samples, SCP2 has 200 samples), which increases the risk of overfitting. In such low-data regimes, models often exhibit low training error (low bias) but high variance, making them prone to overconfidence and instability. This instability is reflected in the geometry of the loss landscape: models may converge to sharp minima, where small perturbations in input or parameters cause large changes in loss. Sharpness thus captures this sensitivity, and sharp minima are empirically linked to poor generalization.
>
> Additionally, time series tasks frequently involve domain drift, a shift between the distribution of training and test data, due to factors like temporal changes or varying sensor conditions. Even when the dataset is moderately sized, this distributional shift can exacerbate the generalization gap. Encouraging the model to find flatter minima, i.e. correspond to solutions that are less sensitive to such shifts, can improve robustness and performance on unseen data.
>
> ---
>
> ### **Q2. Analysis of Epsilon.**
>
> The sensitivity analysis of $\epsilon$ is presented in Section 6 and Fig. 5. While some datasets exhibit performance fluctuations, setting $\epsilon = 2$ provides a generally reasonable balance. To be more comprehensive, we have extended the analysis to include all 26 datasets (consistent with Table 1).
>
> |$\epsilon$|0.02|0.5|2|4|20|100|
> |-|-|-|-|-|-|-|
> |Acc.|69.7|75.2|76.9|76.2|75.5|72.6|
>
> We observe that when $\epsilon$ is set too high, the gain is marginal. This is due to, in this case, the fisher information of the network is very likely below  $\epsilon$, and accordingly, we do activate the renormalization mechanism. Conversely, if $\epsilon$ is too small, model updates become challenging because we always renormalize its gradient. This may lead to performance degradation.
>
> The optimal choice of $\epsilon$ depends on both the dataset and model architecture. While $\epsilon = 2$ serves as a strong default, we also report the results obtained using the best-matched $\epsilon$ for each individual dataset.
>
> |Uniform $\epsilon$| Best-matched  $\epsilon$|
> |-|-|
> |76.2|**77.5**|
>
>
> For a new dataset, $\epsilon$ should be treated as a hyperparameter. However, **its search space can be efficiently narrowed by considering the initial range suggested by the Fisher information estimated during the first few iterations.**
>
> ---
>
> ### **Q3. More Comparison.**
> We compare our method with NuTime (29 datasets selected by NuTime) as follows. SAN is a variant of RevIN that may have similar issues to RevIN, while NuTime is a self-supervised method. Our method demonstrates better overall performance than both of them.
> **Full comparison is available at https://github.com/AnonymousUserss/ICML2025-4119-Response.**
>
> ||Ours (UNI.)|NuTime (self-supervised)|ITIME+SAN|
> |-|:-:|:-:|:-:|
> |Avg.|**78.3**|77.8|74.4|
>
> We will include it in our final version.
>
> ---
>
> ### **Q4. Rationality of the diagonal approximation.**
> We argue that this is a necessary trade-off, i.e., sacrificing the precision of FIM to achieve a feasible computational cost. As mentioned in G.4, we have tested computing the full FIM with a mini-batch size of 64 in **one iteration** on InceptionTime ($\sim$ 0.6M parameters) on an A100 40GB GPU, and it has been running for more than 15 minutes and is still ongoing. We realized that even in such a small-scale network, the computation required for one iteration is prohibitively heavy, and we would need thousands of iterations for full training; hence, applying full FIM is infeasible in broader scenarios with larger networks and/or larger-scale datasets.
> - We also found related works that consistently apply diagonal approximation to tackle a similar computational issue, and they mention that the diagonal elements contain sufficient important information.
>
> We will include these discussions in our final version, but indeed, this is a potential limitation of our work, and we plan to investigate it in the future.
>
> [1] Overcoming catastrophic forgetting in neural networks
>
> [2] Overcoming Catastrophic Forgetting by Incremental Moment Matching
>
> [3] Fedfisher: Leveraging fisher information for one-shot federated learning
>
> ---
>
> ### **Q5. How FIC works.**
>
> In FIC-TSC, whenever the Fisher Information exceeds some threshold $\epsilon$, the parameter gradients are downscaled accordingly. By “capping” how large Fisher information can grow, the algorithm actively steers the model away from sharp or overly sensitive parameter regions and potentially reduces the overall Fisher information.
>
> ---
>
> ### **Q6. Grid Search.**
>
> Following TsLaNet and ConvTran, we split 20\% of data from the training set as a validation set.

---

> > ### Comment · Reviewer_kJBN · 2025-04-04
> >
> > The authors' replies address most of my concerns, and I hold a generally positive view of the paper. However, I consider a score of 3 to be appropriate and do not intend to revise it. While the proposed method is supported by a coherent motivation, the paper lacks model design considerations specific to time-series data characteristics. Lastly, two minor points remain to be addressed:
> >
> > **Q4**: Concern solved. Please include the mentioned relevant references in the paper.
> >
> > **Q5**: Reporting only averaged accuracy is insufficient. With a large number of datasets, extremely high or low values can skew the average. For instance, based on results from the authors’ shared anonymous link, NuTime shows a better average rank than UNI on UEA 29 datasets. Also, the p-value suggests no significant difference between the two. Thus, highlighting only UNI’s higher averaged accuracy offers limited insight.
> >
> >
> > | Dataset                   | UNI   | NuTime |
> > |---------------------------|-------|--------|
> > | ArticularyWordRecognition| 99.3  | 99.4   |
> > | AtrialFibrillation        | 56.7  | 34.7   |
> > | BasicMotions              | 100   | 100    |
> > | CharacterTrajectories     | 99.7  | 99.4   |
> > | Cricket                   | 100   | 100    |
> > | DuckDuckGeese             | 65.0  | 55.2   |
> > | EigenWorms                | 85.5  | 91.0   |
> > | ERing                     | 91.9  | 98.6   |
> > | Epilepsy                  | 98.9  | 99.3   |
> > | EthanolConcentration      | 39.2  | 46.6   |
> > | FaceDetection             | 68.4  | 66.3   |
> > | FingerMovements           | 65.0  | 61.2   |
> > | HandMovementDirection     | 49.3  | 53.2   |
> > | Handwriting               | 61.6  | 22.8   |
> > | Heartbeat                 | 81.0  | 78.4   |
> > | JapaneseVowels            | 99.1  | 98.3   |
> > | Libras                    | 79.4  | 97.6   |
> > | LSST                      | 65.3  | 69.3   |
> > | MotorImagery              | 65.0  | 62.2   |
> > | NATOPS                    | 98.9  | 94.0   |
> > | PEMS-SF                   | 79.2  | 92.5   |
> > | PenDigits                 | 97.6  | 98.8   |
> > | PhonemeSpectra            | 31.3  | 32.0   |
> > | RacketSports              | 89.8  | 93.4   |
> > | SelfRegulationSCP1        | 90.1  | 89.9   |
> > | SelfRegulationSCP2        | 59.4  | 60.3   |
> > | StandWalkJump             | 63.3  | 66.7   |
> > | SpokenArabicDigits        | 100   | 99.3   |
> > | UWaveGestureLibrary       | 90.2  | 95.5   |
> > | **Average Rank**          | 1.52  | **1.41** (small is better)   |
> > | **P-value**               |       | 0.3981 |

---

> > > ### Author Response · Authors · 2025-04-04
> > >
> > > **We sincerely extend our gratitude to the reviewer for the thoughtful comments and the effort in the response. We greatly appreciate your generally positive view of our paper and your acknowledgment that most concerns have been addressed.** Below, we address the two remaining minor points:
> > >
> > > ---
> > >
> > >  **Model design considerations specific to time-series data characteristics.**
> > >
> > > We thank the reviewer for pointing this out. While our method primarily focuses on addressing domain shift in general time series data via Fisher Information Constraint, we agree that explicit modeling of time-series characteristics such as temporal locality, seasonality, or autocorrelation could further enhance performance. In future work, we plan to explore integrating time-series-specific inductive biases (e.g., temporal convolutions or frequency-domain features) into our framework while maintaining the FIC regularization for improved generalizability.
> > >
> > > ---
> > >
> > > **Include the mentioned relevant references in the paper.**
> > >
> > >  We will definitely incorporate the relevant references as suggested into the final version of the paper to ensure proper attribution and contextual completeness.
> > >
> > > ---
> > >
> > > **Compare with NuTime.**
> > >
> > > To better reflect the relationship between UNI and NuTime, we will revise our wording in the manuscript to explicitly state that out method (UNI.) is **on a par with** NuTime, rather than implying any definitive superiority. This adjustment will more accurately convey the results in light of the average accuracy, average rank, and p-value.
> > >
> > >
> > > ---
> > >
> > > ## We thank you again for the time and effort, and please let us know if you have any further questions.

---

### Decision · Program_Chairs · 2025-05-01

**Decision:**

Accept (poster)

**Comment:**

This paper addresses the problem of time-series classification under distribution shift. Its main contribution is the proposed FIC-TCS method, which leverages a Fisher information constraint to guide the model toward flatter minima, thereby improving robustness. During the rebuttal phase, reviewers raised concerns about the motivation related to sharpness in time-series data and the assumption that flatter minima lead to better generalization. These concerns were adequately addressed by the authors. Following the rebuttal, all reviewers gave positive evaluations. The authors are encouraged to incorporate the reviewers’ constructive feedback into the revised manuscript.